# Outsourcing Training without Uploading Data via Efficient Collaborative Open-Source Sampling

**Junyuan Hong**[*]
Michigan State University
hongju12@msu.edu

**Lingjuan Lyu**
Sony AI
lingjuan.lv@sony.com

**Jiayu Zhou**
Michigan State University
jiayuz@msu.edu

**Michael Spranger**
Sony AI
michael.spranger@sony.com

## Abstract

As deep learning blooms with growing demand for computation and data resources, outsourcing model training to a powerful cloud server becomes an attractive alternative to training at a low-power and cost-effective end device. Traditional outsourcing requires uploading device data to the cloud server, which can be infeasible in many real-world applications due to the often sensitive nature of the collected data and the limited communication bandwidth. To tackle these challenges, we propose to leverage widely available *open-source data*, which is a massive dataset collected from public and heterogeneous sources (e.g., Internet images). We develop a novel strategy called Efficient Collaborative Open-source Sampling (ECOS) to construct a proximal proxy dataset from open-source data for cloud training, in lieu of client data. ECOS probes open-source data on the cloud server to sense the distribution of client data via a communication- and computation-efficient sampling process, which only communicates a few compressed public features and client scalar responses. Extensive empirical studies show that the proposed ECOS improves the quality of automated client labeling, model compression, and label outsourcing when applied in various learning scenarios.

## 1 Introduction

Nowadays, powerful machine learning services are essential in many devices that supports our daily routines. Delivering such services is typically done through client devices that are power-efficient and thus very restricted in computing capacity. The client devices can collect data through built-in sensors and make predictions by machine learning models. However, their stringent computing power often makes the local training prohibitive, especially for high-capacity deep models. One widely adopted solution is to outsource the cumbersome training to cloud servers equipped with massive computational power, using machine-learning-as-a-service (MLaaS). Amazon Sagemaker [29], Google ML Engine [6], and Microsoft Azure ML Studio [4] are among the most successful industrial adoptions, where users upload training data to designated cloud storage, and the optimized machine learning engines then handle the training. One major challenge of the outsourcing solution in many applications is that the local data collected are sensitive and protected by regulations, therefore prohibiting data sharing. Notable examples include General Data Protection Regulation (GDPR) [1] and Health Insurance Portability and Accountability Act (HIPPA) [2].

---

[*]Work done during internship at Sony AI. Corresponding to: Lingjuan Lyu.

36th Conference on Neural Information Processing Systems (NeurIPS 2022).

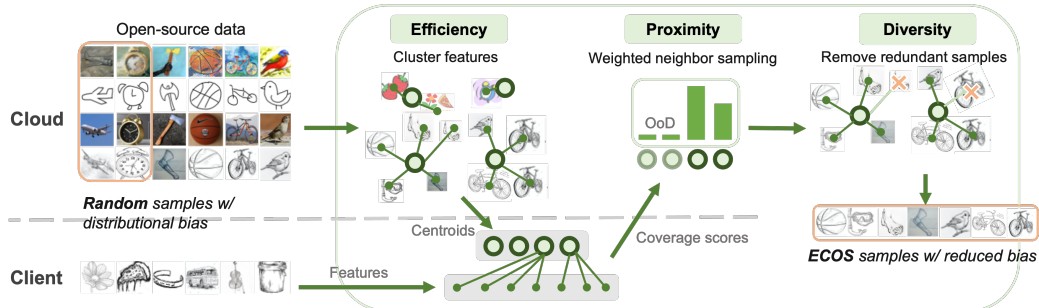

Figure 1: Illustration of the proposed ECOS framework. Instead of uploading local data for cloud training, ECOS downloads the centroids of clustered open-source features to *efficiently* sense the client distribution, where the client counts the local neighbor samples of the centroids as the coverage score. Based on the the scores of centroids, the server adaptively samples *proximal* and *diverse* data for training a transferable model on the cloud.

On the other hand, recent years witnessed a surging amount of general-purpose and massive datasets authorized for public use, such as ImageNet [15], CelebA [31], and MIMIC [26]. Moreover, many task-specific datasets used by local clients can be well considered as biased subsets of these large public datasets [41, 28]. Therefore, the availability of these datasets allows us to use them to model confidential local data, facilitating training outsourcing without directly sharing the local data. One approach is to use the private client dataset to craft pseudo labels for a public dataset in a confidential manner [56, 40], assuming that the public and local data are identically-and-independently-distributed (*iid*). In addition, Alon *et al.* showed that an *iid* public data can strongly supplement client learning, which greatly reduces the private sample complexity [3]. However, the *iid* assumption can often be too strong for general-purpose *open-source* datasets, since they are usually collected from heterogeneous sources with distributional biases from varying environments. For example, a search of 'digits' online yields digits images from handwriting scans, photos, to artwork of digits.

In this paper, we relax the *iid* assumption in training outsourcing and instead consider the availability of an open-source dataset. We first study the gap between the *iid* data and the heterogeneous open-source data in training outsourcing, and show the low sample efficiency of open-source data. We show that in order to effectively train a model from open-source data that is transferable to the client data, the open-source data needs to communicate more samples than those of *iid* data. The main reason behind such low sample efficiency is that we accidentally included out-of-distribution (OoD) samples, which poison the training and significantly degrade accuracy at the target (client) data distribution [5]. We propose a novel framework called Efficient Collaborative Open-source Sampling (ECOS) to tackle this challenge, which filters the open-source dataset through an efficient collaboration between the client and server and does not require client data to be shared. During the collaboration, the server sends compressed representative features (centroids) of the open-source dataset to the client. The client then identifies and excludes OoD centroids and returns their privately computed categorical scores to the server. The server then adaptively and diversely decompresses the neighbors of the selected centroids. The main idea is illustrated in Fig. 1.

Our major contributions are summarized as follows:
• *New problem and insight*: Motivated by the strong demands for efficient and confidential outsourcing, using public data in place of the client data is an attractive solution. However, the impact of heterogeneous sources of the public data, namely open-source data, is rarely studied in existing works. Our empirical study shows the potential challenges due to such heterogeneity.
• *New sampling paradigm*: We propose a new unified sampling paradigm, where the server only sends very few query data to the client and requests very few responses that efficiently and privately guide the cloud for various learning settings on open-source data. To our best knowledge, our method enables efficient cloud outsourcing under the most practical assumption of open-source public data, and does not require accessing raw client data or executing cumbersome local training.
• *Compelling results*: In all three practical learning scenarios, our method improves the model accuracy with pseudo, manual or pre-trained supervisions. Besides, our method shows competitive efficiency in terms of both communication and computation.

## 2 Related Work

There are a series of efforts studying how to leverage the data and computation resources on the cloud to assist client model training, especially when client data cannot be shared [53, 48]. We categorize them as follows: *1) Feature sharing*: Methods like group knowledge transfer [21], split learning [47] and domain adaptation [18, 17] transfer edge knowledge by communicating features extracted by networks. To provide a theoretical guarantee of privacy protection, [37] proposed an advanced information removal to disentangle sensitive attributes from shared features. In the notion of rigorous privacy definition, Liu *et al.* leveraged public data to assist private information release [30]. Earlier, data encryption was used for outsourcing, which however is too computation-intensive for a client and less applicable for large-scale data and deep networks [12, 27]. Federated Learning (FL) [34] considers the same constraint on data sharing but allocates the burdens of training [23] and communication [57] to clients and opens up a series of challenges on privacy [11], security [33, 10] and knowledge transfer [24]. *2) Private labeling:* PATE and its variants were proposed to generate client-approximated labels for unlabeled public data, on which a model can be trained [39, 40]. Without training multiple models by clients, Private kNN was a more efficient alternative which explored the private neighborhood of public images for labeling [56]. These approaches are based on a strong assumption of the availability of public data that is *iid* as the local data. This paper considers a more practical yet challenging setting where public data are from multiple agnostic sources with heterogeneous features.

Sampling from public data has been explored in central settings. For example, Xu *et al.* [51] used a few target-domain samples as a seed dataset to filter the open-domain datasets by positive-unlabeled learning [32]. Yan *et al.* [52] used a model to find the proxy datasets from multiple candidate datasets. In self-supervised contrastive learning, model-aware $K$-center (MAK) used a model pre-trained on the seed dataset to find desired-class samples from open-world dataset [25]. Though these methods provided effective sampling, they are less applicable when the seed dataset is placed at the low-energy edge, because the private seed data at the edge cannot be shared with the cloud for filtering and the edge device is incapable of computation-intensive training. To address these challenges, we develop a new sampling strategy requiring only light-weight computation at the edge.

## 3 Outsourcing Model Training With Open-Source Data

### 3.1 Problem Setting and Challenges

Motivated in Section 1, we aim to outsource the training process from computation-constrained devices to the powerful cloud server, where a proxy public dataset without privacy concerns is used in place of the client dataset for cloud training. One solution is (private) client labeling by k-nearest-neighbors (kNN) [56], where the client and cloud server communicate the pseudo-label of a public dataset privately and the server trains a classifier by the labeled and unlabeled samples in a semi-supervised manner. The success of this strategy depends on the key assumptions that public data in the cloud and private data in the client are *iid*, which are rather strong in practice and thus prevent it from many real-world applications. In this work, we make a more *realistic* assumption that the public datasets are as accessible as *open-source* data. An open-source dataset consists of

Table 1: Test accuracy (%) with different client domains (columns). Cloud data are identically distributed as the client data (*ID*) or including more data from 5 distinct domains (*ID+OoD*) without overlapped samples. We first label a number of randomly selected cloud examples (i.e., sampling budget) privately by client data [56], and then train a classifier to recognize digit images. The privacy cost $\epsilon$ is accounted for in the notion of differential privacy. Larger budgets imply more privacy and communication costs. More results on different settings are enclosed in Appendix B.3.

| Cloud Data | Sampling Budget | MNIST Acc (%) ↑ | $\epsilon$ ↓ | SVHN Acc (%) ↑ | $\epsilon$ ↓ | USPS Acc (%) ↑ | $\epsilon$ ↓ | SynthDigits Acc (%) ↑ | $\epsilon$ ↓ | MNIST-M Acc (%) ↑ | $\epsilon$ ↓ | Average Acc (%) ↑ | $\epsilon$ ↓ |
|---|---|---|---|---|---|---|---|---|---|---|---|---|---|
| ID | 1000 | **84.3**±2.4 | 4.48 | **51.6**±1.4 | 4.08 | **87.1**±0.5 | 4.51 | **73.2**±1.5 | 4.57 | **55.5**±1.0 | 4.46 | **70.4** | 4.42 |
| ID+OoD | 1000 | 78.0±3.5 | 4.30 | 40.6±1.6 | 3.75 | 82.2±2.7 | 4.32 | 62.1±1.6 | 4.41 | 49.1±1.0 | 4.27 | 62.4 | 4.21 |
| | 8000 | 82.2±4.1 | 5.89 | 47.9±1.8 | 5.89 | 85.4±0.5 | 5.89 | 64.4±3.6 | 5.89 | 53.3±2.2 | 5.89 | 66.6 | 5.89 |
| | 16000 | 82.6±1.4 | 7.17 | 48.5±1.7 | 7.17 | 86.7±1.9 | 7.17 | 67.5±2.3 | 7.17 | 52.0±3.0 | 7.17 | 67.4 | 7.17 |
| | 32000 | 84.1±1.6 | 9.32 | 49.4±0.2 | 9.32 | 86.8±2.0 | 9.32 | 68.5±0.1 | 9.32 | 53.0±2.7 | 9.32 | 68.4 | 9.32 |

biased features from multiple heterogeneous sources (feature domains), and therefore includes not only in-distribution (ID) samples similar to the client data but also multi-domain OoD samples.

The immediate question is how the OoD samples affect the outsourced training. In Table 1, we empirically study the problem by using a 5-domain dataset, Digits, where 50% of one domain is used on the client and the remained 50% together with the other 4 domains serve as the public dataset on the cloud. To conduct the cloud training, we leverage the client data to generate pseudo labels for the unlabeled public samples, following [56]. It turns out that the presence of OoD samples in the cloud greatly degrades the model accuracy. The inherent reason for the degradation is that the distributional shift of data [43] compromised the transferability of the model to the client data [50].

**Problem formulation by sampling principles**. Given a client dataset $D^p$ and an open-source dataset $D^q$, the goal of open-source sampling is to find a proper subset $S$ from $D^q$, whose distribution matches $D^p$. In [25], Model-Aware K-center (MAK) formulated the sampling as a principled optimization:

$$\min_{S \subseteq D^q} \Delta(S, D^p) - H(S \cup D^p; D^q), \tag{1}$$

where $\Delta(S, D^p) := \mathbb{E}_{x' \in D^p}[\min_{x \in S} \|\phi(x) - \phi(x')\|^2]$ measures *proximity* as the set difference between $S$ and $D^p$ using a feature extractor $\phi$, and the latter $H(S \cup D^p; D^q) := \max_{x' \in D^q} \min_{x \in S \cup D^p} \|\phi(x) - \phi(x')\|^2$ measures *diversity* by contradicting $S \cup D^p$ and $D^q$ (suppose $D^q$ is the most diverse set)[2]. Solving Eq. (1) results in an NP-hard problem that is intractable [13], and MAK provides an approximated solution by a coordinate-wise greedy strategy. It first pre-trains the model representations on $D^p$ and finds a large candidate set with the best proximity to extracted features. Then, it incrementally selects the most diverse samples from the candidate set until the sampling budget is used up.

Though MAK is successful in the central setting, it is not applicable when $D^p$ is isolated from cloud open-source data and is located at a resource-constrained client for two reasons: 1) *Communication inefficiency*. Uploading client data may result in privacy leakage, sending public data to the client is a direct alternative but the cost can be prohibitive. 2) *Computation inefficiency*. Pre-training a model on $D^p$ or proximal sampling (which computes the distances between paired samples from $D^q$ and $D^p$) induces unaffordable computation overheads for the low-energy client.

### 3.2 Proposed Solution: Efficient Collaborative Open-Source Sampling (ECOS)

To address the above challenges, we design a new strategy that 1) uses compressed queries to reduce the communication and computation overhead and 2) uses a novel principled objective to effectively *sample* from open-source data with the client responses of the compressed queries.

**Construct communication-efficient and an informative query set $\hat{\Phi}^q$ at cloud**. Let $d$ be the number of pixels of an image, the communication overhead of transmitting $D^q$ to the client is given by $\mathcal{O}(d|D^q|)$. For communication efficiency, we optimize the following two factors:
i) *Data dimension $d$*. First, we transmit extracted features $\Phi^q = \{\phi(x)|x \in D^q\}$ instead of images to reduce the communication overhead to $\mathcal{O}(d_e|D^q|)$, where $d_e$ is a much smaller embedding dimension. For accurate estimation of the distance $\Delta$, a pre-defined discriminative feature space is essential without extra training on the client. Depending on resources, one may consider hand-crafted features such as HOG [14], or an off-the-shelf pre-trained model such as ResNet pre-trained on ImageNet.
ii) *Data size $|D^q|$*. Even with compression, sending all data for querying is inefficient due to the huge size of open-source data $|D^q|$. Meanwhile, too many queries would cast unacceptable privacy costs to the client. As querying on similar samples leads to redundant information in querying, we propose to reduce such redundancy by selecting informative samples. We use the classic clustering method KMeans [20] for compressing similar samples by clustering them, and collect the $R$ mean vectors or *centroids* into $\hat{\Phi}^q = \{c_r\}_{r=1}^R$. We denote $R$ as the *compression size* and $\hat{D}^q$ as the set of original samples corresponding to $\hat{\Phi}^q$.

**New sampling objective**. We note that sending the compact set $\hat{\Phi}^q$ in place of $D^q$ prohibits the client from optimizing $\Delta(S, D^p)$ in Eq. (1) for $S \in D^q$. Instead, we sample a set of centroids $\hat{S} \in \hat{\Phi}^q$ and decompress them by the cluster assignment into corresponding original samples with rich features

---

[2]Note that we use $L_2$-norm distance instead of normalized cosine similarity in $\Delta(S, D^p)$ in contrast to MAK, since normalized cosine similarity is not essential if the feature space is not trained under the cosine metric. We also omit the tailedness objective which is irrelevant in our context.

afterwards. In principle, we leverage the inequality $\Delta(S, D^p) \leq \Delta(\hat{S}, D^p) + \Delta(\hat{S}, S)$ to attain a communication-efficient surrogate objective as follows:

$$\min_{\hat{S} \subseteq \hat{D}^q, S \subseteq D^q} \underbrace{\Delta(\hat{S}, D^p) + \Delta(\hat{S}, S)}_{\text{proximity}} - \underbrace{H(S; D^q)}_{\text{diversity}}, \tag{2}$$

where $\hat{S}$ (or $\hat{D}^q$) is the compressed centroid substitute of $S$ (or $D^q$). Different from Eq. (1), we decompose the proximity term into two in order to facilitate communication efficiency leveraging an informative subset $\hat{D}^q$. We solve the optimization problem in a greedy manner by two steps at the client and the cloud, respectively:

i) At the *client step*, we optimize $\Delta(\hat{S}, D^p)$ to find a subset of centroids ($\hat{S} \subset \hat{D}^q$) that are proximal to the client set $D^p$. Noticing that $\hat{D}^q$ contains the cluster centroids, we take advantage of the property to define a novel proximity measure of the cluster $r$: Centroid Coverage (CC), denoted as $v_r$. Upon receiving centroids from the cloud, the client uses them to partition the local data into $\{\mathcal{C}_r^p\}_{r=1}^R$ where $\mathcal{C}_r^p$ denotes the $r$-th cluster partition of local data. We compute the CC score by the cardinality of the neighbor samples of the centroid $r$, i.e., $v_r = |\mathcal{C}_r^p|$. To augment the sensitivity to the proximal clusters, we scale the CC score by a non-linear function $v_r' = \psi_s(|\mathcal{C}_r^p|)$, where the scale function $\psi_s(x) = x^s$ is parameterized by $s$.

ii) At the *cloud step*, we optimize the proximity of $S$ w.r.t. the proxy set $\hat{S}$, i.e., $\Delta(\hat{S}, S)$, and remove redundant and irrelevant samples from the candidate set to encourage diversity, i.e., $-H(S; D^q)$. As samples among clusters are already diversified by KMeans, we only need to promote the in-cluster *diversity*. To this end, we reduce the sample redundancy per cluster at cloud by K-Center [44], which heuristically finds the most diverse samples. Such design transfers the diversity operation to cloud and thus reduces the local computation overhead. To maintain the *proximity*, the K-Center is applied within each cloud cluster and the sampling budget per cluster is proportional to their vote numbers and the original cluster sizes. With the normalized scores, we compute the sampling budget per cluster which is upper bounded by the ratio of the cluster in the cloud set.

---

**Algorithm 1** Efficient collaborative open-source sampling (ECOS)

---

**Input:** Client dataset $D^p$, cloud query dataset $D^q$, sampling budget $B$, compression size $R$, feature extractor $\phi(\cdot)$, distance function $\Delta(x, S) = \min_{y \in S} \|\phi(x) - \phi(y)\|$, initial sample set $S = \emptyset$, score scale function $\psi_s(x) = x^s$.

1: Extract features $\Phi^q = \{\phi(x) | x \in D^q\}$;
2: Cloud creates a compressed dataset $\hat{\Phi}^q = \text{KMeans}_R(\Phi^q)$;          ▷ Compress $R$ Centroids
3: ▷▷▷ **Client End** ▷▷▷
4: Download the feature extractor $\phi$ and $\hat{\Phi}^q$;
5: Use centroids $\hat{\Phi}^q$ to partition $\Phi^p = \{\phi(x) | x \in D^p\}$ into clusters $\{\mathcal{C}_r^p\}_{r=1}^R$;
6: Compute the Centroid Coverage (CC) scores: $v_r = |\mathcal{C}_r^p|, \forall r \in \{1, \ldots, R\}$;
7: Upload scaled cluster scores $\{v_r' = \psi_s(v_r)\}_{r=1}^R$;          ▷ Proximity
8: ◁◁◁ **Cloud End** ◁◁◁
9: Partition $D^q$ into clusters $\{\mathcal{C}_r^q\}_{r=1}^R$ by centroids $\hat{\Phi}^q$;
10: Compute per-cluster sampling budget $b_r = \min\left\{\frac{|\mathcal{C}_r^q|}{\sum_j |\mathcal{C}_j^q|}, \frac{v_r'}{\sum_j v_j'}\right\} \cdot B$;
11: **for** $r$ in $\{1, \ldots, R\}$ **do**          ▷ Decompress Centroids
12:     Initialize $S' = \{x\}$ by a sample randomly picked from $\mathcal{C}_r$;
13:     **while** $|S'| < b_r$ **do**          ▷ Diverse Sampling
14:         $u = \arg\max_{x \in \mathcal{C}_r^q} \Delta(x, S')$;
15:         $S' = \{u\} \cup S'$;
16:     $S = S' \cup S$;
17: **return** $S$

---

We summarize our algorithm in Algorithm 1, which enables the clients to enjoy better computation efficiency than local training and better communication efficiency than the centralized sampling (e.g., MAK). **1) Computation efficiency**. Since our method only requires inference operations on the client device, which should be efficiently designed for the standard predictive functions of the device, and is training-free for the client, the major complexity of ECOS is on computing centroid coverage and is much lower than gradient-based algorithms whose complexity scales with the model size and training

iterations. As computing the CC scores only requires the nearest centroid estimation and ranking, the filtering can be efficiently done. The total *time complexity* is $\mathcal{O}(C_\phi|D^p| + (d_e + 1)R|D^p|)$, dominated by the first term, where $C_\phi$ is the complexity of extracting features depending on the specific method. The second term $(d_e + 1)R|D^p|$ is for computing the pair-wise distances between $\Phi^p$ and $\hat{\Phi}^q$ and estimating the nearest centroids per sample (or partitioning client data). In a brief comparison, the complexity of local $T$-iteration gradient-descent training could be approximately $\mathcal{O}(TC_\phi|D^p|)$ which is much more expensive since typically $TC_\phi \gg d_e$. To complete the analysis, the *space complexity* is $\mathcal{O}(C'_\phi + (d + d_e)|D^p| + d_eR + |D^p|R)$ for the memory footprint of $\phi$, the images ($d$) or features ($d_e$) of client and cloud centroid data, and the distance matrix. **2) Communication efficiency**. The downloading complexity will be $\mathcal{O}(d_eR)$ for $R$ $d_e$-dimensional centroid features and the uploading complexity is $\mathcal{O}(R)$ including indexes of samples. Thus, the data that will be communicated between the client and cloud is approximately of $\mathcal{O}(d_eR + R)$ complexity in total. In comparison, downloading the whole open-source dataset by central sampling (e.g., MAK) requires $\mathcal{O}(d|D^q| + B)$ complexity. As $R < d_eR \ll d_e|D^q| \ll d|D^q|$, our method can significantly scale down the computation cost.

**Privacy protection and accountant**. When the cloud server is compromised by an attacker, uploading CC scores leak private information of the local data samples, for example, the presence of an identity [45]. To mitigate the privacy risk, we protect the uploaded scores by a Gaussian noise mechanism, i.e., $\tilde{v}_r = v_r + \mathcal{N}(0, \sigma^2)$, and account for the privacy cost in the notion of differential privacy (DP) [19]. DP quantifies the numerical influence of the absence of a private sample on $[v_1, \cdots, v_R]$, which is connected to the chance of exposing the sample to the attacker. To obtain a tight bound on the privacy cost, we utilize the tool of Rényi Differential Privacy (RDP) [36] and leverage the Poisson sampling to further amplify the privacy [55]. With the noise mechanism governed by $\sigma$, the resultant privacy cost in the sense of $(\epsilon, \delta)$-DP can be accounted as $\epsilon = \mathcal{O}(\gamma\sqrt{\log(1/\delta)}/\sigma)$ where $\gamma$ is the Poisson subsampling rate and $\delta$ is a user-specified parameter. A larger $\epsilon$ implies higher risks of privacy leakage in the probability of $\delta$. Formal proofs can be found in Appendix A.3.

**Generalization error of models trained on cloud.** Our work can be viewed as knowledge transfer from the open-source domain to the private client domain. Therefore, we present Theorem 3.1 based on prior domain-adaptation theoretical results [7, 18, 17].

**Theorem 3.1.** *Assume that a open-source dataset $S$ is induced from a mixture of cluster distributions, i.e., $\sum_{r=1}^R \alpha_r \mathcal{D}_r^q$ with cluster distribution $\mathcal{D}_r^q$, $\alpha_r \in [0, 1]$ and $\sum_{r=1}^R \alpha_r = 1$. Suppose client data are sampled from $\mathcal{D}^p$. Let $L(\cdot, \cdot)$ be a loss function on a hypothesis and a dataset (for empirical error) or a distribution (for generalization error). If $f$ is governed by the parameter $\theta$ trained on $S$ and belongs to a hypothesis space $\mathcal{H}$ of $VC$-dimension $d$, then with probability at least $1 - p$ over the choice of samples, the inequality holds,*

$$L(f_\theta, \mathcal{D}^p) \le L(f_\theta, S) + \frac{1}{2}\sum_{r=1}^R \alpha_r d_{\mathcal{H}\Delta\mathcal{H}}(\mathcal{D}_r^q, \mathcal{D}^p) + 4\sqrt{\frac{2d\log(2|S|) + \log(4/p)}{|S|}} + \xi, \quad (3)$$

*where $\xi = \min_\theta \{L(f_\theta, S) + L(f_\theta, D^p)\}$, and $d_{\mathcal{H}\Delta\mathcal{H}}(\mathcal{D}, \mathcal{D}')$ denotes the distribution divergence.*

The proof of Theorem 3.1 is deferred to Appendix A.4. Theorem 3.1 shows that given a model trained on cloud, its generalization error on client data hinges on the quality and size of the ECOS-sampled subset. Informally, if the sampled set has enough samples (the 3rd term) and follows a similar distribution as the client dataset (the 2nd term), then the model generalizes better via only training on the proxy cloud dataset (the 1st term). In Theorem 3.1, the model $f_\theta$ can be trained by any task-specific optimization method. For example, minimizing a $\mu$-strongly-convex and $G$-smooth loss function $L(f_\theta, S)$ w.r.t. $\theta$ via $T$-iteration gradient descent leads to $L(f_\theta, S) = (1 - \mu/G)^T|L(f_{\theta_0}, S) - L^*|$ where $L^*$ is the optimal loss.

Given Eq. (3), it can be shown that minimizing the upper bound requires a dedicated trade-off between sample size and quality. Consider a simple case, only minimizing $\sum_{r=1}^R \alpha_r d_{\mathcal{H}\Delta\mathcal{H}}(\mathcal{D}_r^q, \mathcal{D}^p)$. The solution is that $\alpha_r$ equals 1 if $d_{\mathcal{H}\Delta\mathcal{H}}(\mathcal{D}_r^q, \mathcal{D}^p)$ is smallest among all choices of clusters, otherwise zero. However, the number of samples in the cluster $r$ is limited as $|\mathcal{C}_r^q|$ which is much smaller than the whole open-source dataset $|D^q|$, and therefore $|S|$ will be too small to enlarge the third term in Eq. (3). Thus, $\alpha_r$ should be smoothed to increase sample size by trading in some quality (distribution divergence). In our implementation, we approximate $\alpha_r$ via the CC score $v_r^s$, where an $s < \infty$ smooths the $\alpha_r$ to trade off sample size and quality. When $s$ vanishes, all clusters will be selected at the same chance and the divergence could be large.

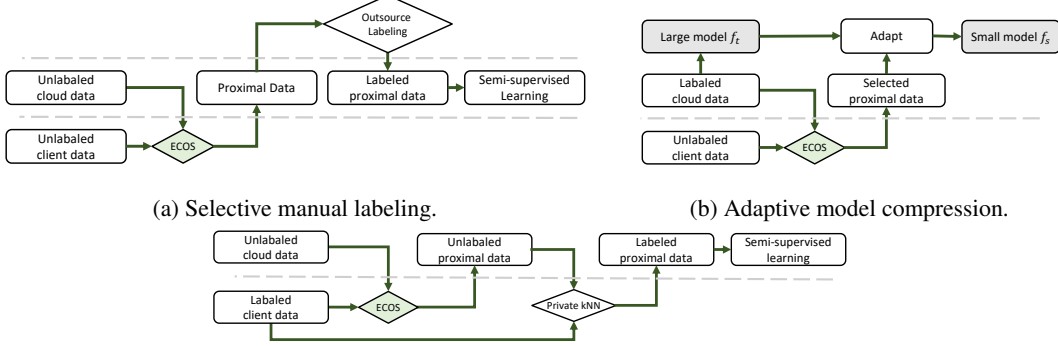

(a) Selective manual labeling.      (b) Adaptive model compression.

(c) Automated client labeling.

Figure 2: Our method is applicable to various cloud training cases, where ECOS filters the open-source samples to improve the model performance trained on (a) manual, (b) pre-trained model (teacher), and (c) pseudo supervisions.

## 4 Empirical Results

**Datasets.** We use datasets from two tasks: digit recognition and object recognition. Distinct from prior work [56], in our work, the open-source data contains samples out of the client's distribution. With the same classes as the client dataset, we assume open-source data are from different environments and therefore include different feature distributions, for example, DomainNet [41] and Digits [28]. DomainNet includes large-sized $244 \times 244$ everyday images on 6 domains and Digits consists of $28 \times 28$ digit images on 5 domains. Instead of using an overly large volume of data from a single domain like [56], we tailor each-domain subset to contain fewer images than standard digit datasets, for instance, MNIST with 7438 images, which was previously adopted in the distributed learning setting [28] and mitigates the hardness of collecting enormous data. In practice, collecting tens of thousands of images by a single client, e.g., 50000 images from MNIST domain, will be unrealistic. Similarly, DomainNet will be tailored to only include 10 classes with 2000-5000 images per domain.
**Splits of client and cloud datasets**. For Digits, we use one domain for the client and the rest domains for the cloud as open-source set. For DomainNet, we randomly select 50% samples from one domain for the client and leave the rest samples together with all other domains to the cloud. The difference of configurations for the two datasets is caused by their different domain gaps. Even without ID data, it is possible for Digits to transfer the knowledge across domains.
**Baselines.** For a fair comparison, we compare our method to baselines with the same sampling budget. Each experiment case is repeated for three times with seeds $\{1, 2, 3\}$. We account for the privacy cost by Poisson-subsampling RDP [55] and translate the cost to the general privacy notion, $(\epsilon, \delta)$-DP when $\delta = 10^{-5}$. Here, we use the *random* sampling as a naive baseline. We also adopt a coreset selection method, K-Center [44], to select informative samples within the limited budget. Both baselines are perfectly private, because they do not access private information from clients. Details of hyper-parameters are deferred to Appendix B.1.

### 4.1 Evaluations on Training Outsourcing

To demonstrate the general applicability of ECOS, we present three practical use cases of outsourcing, categorized by the form of supervisions. The conceptual illustrations are in Fig. 2. Per use case, we train a model on partially labeled cloud data (outsourced training) and accuracy on the client test set is reported together with standard deviations. We present the results in Tables 2 to 4 case by case, where we vary the domain of the client by columns. In each column, we highlight the best result unless the difference is not statistically significant.

**Case 1: Selective manual labeling**. We assume that the cloud will label the filtered in-domain samples by using a third-party label service, e.g., Amazon Mechanical Turk [38], or by asking the end clients for manual labeling. As the selected samples are non-private, they can be freely shared with a third parity. But the high cost of manual labeling or service is the pain point, which should be carefully constrained within a finite *budget* of demanded labels.

Table 2: Test accuracy by selective labeling on Digits (top) and DomainNet (bottom).

| Sampling Budget | Method | MNIST Acc (%) ↑ | ε ↓ | SVHN Acc (%) ↑ | ε ↓ | USPS Acc (%) ↑ | ε ↓ | SynthDigits Acc (%) ↑ | ε ↓ | MNIST-M Acc (%) ↑ | ε ↓ | Average Acc (%) ↑ |
|---|---|---|---|---|---|---|---|---|---|---|---|---|
| 2000 | Ours | **97.3**±0.1 | 0.22 | **68.7**±0.3 | 0.22 | 90.8±0.1 | 0.22 | **84.4**±0.6 | 0.22 | 70.4±0.6 | 0.22 | **82.3** |
| | K-Center | 96.7±0.3 | 0.00 | 65.1±1.3 | 0.00 | 90.1±0.7 | 0.00 | 80.2±1.1 | 0.00 | 70.1±0.3 | 0.00 | 80.4 |
| | Random | 96.5±0.3 | 0.00 | 64.0±0.3 | 0.00 | 91.6±1.0 | 0.00 | 83.8±0.3 | 0.00 | 70.9±0.6 | 0.00 | 81.4 |
| 5000 | Ours | **98.1**±0.2 | 0.22 | **74.6**±1.0 | 0.22 | **93.5**±0.3 | 0.22 | **91.2**±0.4 | 0.22 | 74.5±0.5 | 0.22 | **86.4** |
| | K-Center | 97.9±0.2 | 0.00 | 72.3±0.6 | 0.00 | 92.7±0.9 | 0.00 | 89.6±0.3 | 0.00 | 74.0±0.5 | 0.00 | 85.3 |
| | Random | 97.6±0.3 | 0.00 | 70.0±0.3 | 0.00 | 93.0±0.6 | 0.00 | 89.7±0.4 | 0.00 | 73.9±0.7 | 0.00 | 84.8 |
| - | Local | 99.1±0.1 | 0.00 | 88.8±0.2 | 0.00 | 98.9±0.1 | 0.00 | 96.4±0.2 | 0.00 | 88.8±0.2 | 0.00 | 94.4 |

| Sampling Budget | Method | Clipart Acc (%) ↑ | ε ↓ | Infograph Acc (%) ↑ | ε ↓ | Painting Acc (%) ↑ | ε ↓ | Quickdraw Acc (%) ↑ | ε ↓ | Real Acc (%) ↑ | ε ↓ | Sketch Acc (%) ↑ | ε ↓ | Average Acc (%) ↑ |
|---|---|---|---|---|---|---|---|---|---|---|---|---|---|---|
| 1000 | Ours | **88.4**±1.5 | 0.58 | **52.6**±0.9 | 0.58 | **90.4**±1.7 | 0.58 | **84.3**±1.6 | 0.58 | 92.1±1.2 | 0.58 | **87.2**±0.5 | 0.58 | **82.5** |
| | K-Center | 86.8±0.3 | 0.00 | 50.5±0.9 | 0.00 | 89.1±1.4 | 0.00 | 27.2±1.8 | 0.00 | **92.5**±0.1 | 0.00 | 85.6±1.4 | 0.00 | 72.0 |
| | Random | 86.9±0.8 | 0.00 | 47.4±2.7 | 0.00 | 88.6±0.1 | 0.00 | 77.9±2.4 | 0.00 | 91.4±0.3 | 0.00 | 86.9±0.5 | 0.00 | 79.9 |
| 3000 | Ours | 93.2±0.4 | 0.58 | **58.1**±0.6 | 0.58 | 92.5±1.1 | 0.58 | **89.2**±0.9 | 0.58 | **94.4**±0.2 | 0.58 | 92.8±0.2 | 0.58 | **86.7** |
| | K-Center | 93.5±1.1 | 0.00 | 56.3±0.3 | 0.00 | **92.9**±0.3 | 0.00 | 60.5±8.7 | 0.00 | 94.1±0.2 | 0.00 | 92.1±0.7 | 0.00 | 81.6 |
| | Random | 92.5±0.6 | 0.00 | 53.6±1.4 | 0.00 | 91.7±0.8 | 0.00 | 86.1±0.4 | 0.00 | 93.5±0.3 | 0.00 | 93.0±0.2 | 0.00 | 85.1 |
| - | Local | 87.0±0.3 | 0.00 | 51.7±0.7 | 0.00 | 85.9±0.4 | 0.00 | 83.5±0.3 | 0.00 | 93.5±0.1 | 0.00 | 81.7±0.6 | 0.00 | 80.6 |

Given a specified sampling budget, we compare the test accuracy of semi-supervised learning (FixMatch) on sampled data in Table 2. Since the ECOS tends to select in-distribution samples, it eases the transfer of cloud-trained model to the client data. On the Digits dataset, we find that our method attains more accuracy gains as budget increases, demonstrating that more effective labels are selected. On the DomainNet dataset, our method outperforms baselines on 5 out of 6 domains and is stable in most domains given a small budget (1000) and is superior on average. Given higher budgets, the accuracy of all methods are improved, and our method is outstanding on Infograph and Quickdraw domains and is comparable to the baselines on other domains.

Table 3: Test accuracy by adaptive model compression on DomainNet.

| Sampling Budget | Method | Clipart Acc (%) ↑ | ε ↓ | Infograph Acc (%) ↑ | ε ↓ | Painting Acc (%) ↑ | ε ↓ | Quickdraw Acc (%) ↑ | ε ↓ | Real Acc (%) ↑ | ε ↓ | Sketch Acc (%) ↑ | ε ↓ | Average Acc (%) ↑ |
|---|---|---|---|---|---|---|---|---|---|---|---|---|---|---|
| 1000 | Ours | 82.9±0.7 | 0.58 | **48.5**±2.7 | 0.58 | **85.4**±1.0 | 0.58 | 81.3±2.5 | 0.58 | 91.4±0.6 | 0.58 | **82.4**±1.1 | 0.58 | **78.6** |
| | K-Center | 81.2±0.9 | 0.00 | 44.6±0.9 | 0.00 | 84.5±2.2 | 0.00 | 41.7±1.9 | 0.00 | **92.4**±0.5 | 0.00 | 80.1±2.1 | 0.00 | 70.8 |
| | Random | **83.8**±0.7 | 0.00 | 44.4±2.1 | 0.00 | 83.8±1.6 | 0.00 | 76.3±2.2 | 0.00 | 90.1±0.4 | 0.00 | 80.1±0.7 | 0.00 | 76.4 |
| 3000 | Ours | **90.6**±0.6 | 0.58 | **51.4**±1.9 | 0.58 | **89.6**±1.2 | 0.58 | **87.6**±0.3 | 0.58 | 93.6±0.7 | 0.58 | **88.4**±1.5 | 0.58 | **83.5** |
| | K-Center | 88.9±2.3 | 0.00 | 51.2±0.4 | 0.00 | 89.4±0.9 | 0.00 | 57.6±4.4 | 0.00 | **94.5**±0.5 | 0.00 | 86.9±0.6 | 0.00 | 78.1 |
| | Random | 88.4±0.8 | 0.00 | 47.6±1.9 | 0.00 | 89.4±1.0 | 0.00 | 84.7±0.2 | 0.00 | 93.0±1.0 | 0.00 | 86.3±1.2 | 0.00 | 81.6 |

**Case 2: Adaptive model compression**. Due to the large volume of the open-source dataset, a larger model is desired for better capturing the various features, which however is so inefficient to fit into the resource-constrained client devices or specialize for the data distribution of the client. Confronting this challenge, model compression [9] is a conventional idea to forge a memory-efficient model by transferring knowledge from large models to small ones. Specifically, we first pre-train a large *teacher* model $f_t$ on all cloud data by the supervised learning, assuming labels are available in advance. Still, we use an ImageNet-pre-trained model to initialize the feature extractor $\phi(\cdot)$ of a *student* model $f_s$. Then the client will use the downloaded feature extractor $\phi$ to filter samples. Here, we utilize knowledge distillation [22] to finetune the $f_s$ with an additional classifier head upon the $\phi$. On the selected samples, we train a linear classifier head for 30 epochs under the supervision of true labels and the teacher model $f_t$, and then fine-tune the full network $f_s$ for 500 epochs. The major challenge comes from distributional biases between the multi-source open-source data and the client data. Leveraging ECOS, we may sample data near the client distribution and reduce the bias in the follow-up compression process.

We simulate the case on the large-sized image dataset, DomainNet, which is demanding for large-scale networks, e.g., ResNet50, to effectively learn the complicated features. Here, we compress ResNet50 into a smaller network, ResNet18, by using an adaptively selected subset of the cloud dataset. We omit the experiment for digit images where a large model may not be necessary for such a small image size. In Table 3, we present the test accuracy of compressed ResNet18 using 3000 samples from DomainNet in finetuning. With a small portion of privacy cost ($\epsilon < 0.6$), our method improves the accuracy on Clipart and Real domains against the baselines. Note that the model accuracy here is

lower than label outsourcing in Table 2, and reason is that the supervisions from the larger models are just an approximation of the full dataset. Without using the full dataset for compression, the training can be completed fast and responsively on the demand of a client.

Table 4: Test accuracy of client labeling on two datasets: Digits (top) and DomainNet (bottom).

| Sampling Budget | Method | MNIST Acc (%) ↑ | $\epsilon$ ↓ | SVHN Acc (%) ↑ | $\epsilon$ ↓ | USPS Acc (%) ↑ | $\epsilon$ ↓ | SynthDigits Acc (%) ↑ | $\epsilon$ ↓ | MNIST-M Acc (%) ↑ | $\epsilon$ ↓ | Average Acc (%) ↑ |
|---|---|---|---|---|---|---|---|---|---|---|---|---|
| | Ours | **84.2**±2.3 | 5.35 | 47.9±3.1 | 5.32 | **86.1**±1.0 | 5.35 | 68.6±1.6 | 5.35 | **58.4**±1.9 | 5.35 | **69.0** |
| 5000 | K-Center | 81.9±3.4 | 5.34 | 48.4±1.2 | 5.33 | 82.1±1.2 | 5.34 | 69.4±1.9 | 5.34 | 55.4±2.0 | 5.34 | 67.4 |
| | Random | 81.8±4.1 | 5.34 | 45.3±3.0 | 5.29 | 81.2±2.3 | 5.34 | 65.9±2.7 | 5.34 | 55.5±2.6 | 5.34 | 65.9 |

| Sampling Budget | Method | Clipart Acc (%) ↑ | $\epsilon$ ↓ | Infograph Acc (%) ↑ | $\epsilon$ ↓ | Painting Acc (%) ↑ | $\epsilon$ ↓ | Quickdraw Acc (%) ↑ | $\epsilon$ ↓ | Real Acc (%) ↑ | $\epsilon$ ↓ | Sketch Acc (%) ↑ | $\epsilon$ ↓ | Average Acc (%) ↑ |
|---|---|---|---|---|---|---|---|---|---|---|---|---|---|---|
| | Ours | 33.2±5.9 | 4.46 | **23.8**±2.2 | 3.50 | **47.4**±3.3 | 4.51 | **39.8**±7.7 | 4.87 | 62.9±1.9 | 4.92 | **51.7**±1.2 | 4.94 | **43.2** |
| 3000 | K-Center | **39.3**±3.6 | 4.57 | 18.2±2.7 | 3.61 | 46.6±2.5 | 4.52 | 36.1±2.9 | 4.94 | **63.8**±1.9 | 4.96 | 47.7±2.8 | 4.96 | 42.0 |
| | Random | 30.7±2.2 | 4.41 | 21.6±4.9 | 3.43 | 44.0±3.1 | 4.53 | 39.0±5.6 | 4.82 | 59.9±3.6 | 4.75 | 47.2±3.4 | 4.94 | 40.4 |

**Case 3: Automated client labeling**. When the client obtained labeled samples, for example, photos labeled by phone users, the cloud only needs to collect an unlabeled public dataset. Therefore, we may automate the labeling process leveraging the client supervision knowledge to reduce cost or users' efforts on manual labeling. To be specific, we let the client generate pseudo labels for the cloud data based on their neighbor relation, as previously studied by [56] (private kNN). However, the private kNN assumes that the client and the cloud follow the same distribution, which weakens its applicability confronting the heterogeneity and the large scale of open-source data. Thus, we utilize ECOS as a pre-processing of open-source data before being labeled by private kNN. Therefore, we have two rounds of communication for transferring client knowledge: proximal-data sampling by the ECOS and client labeling by the private kNN [56]. To compose the privacy costs from these two steps, we utilize the analytical moment accountant technique to get a tight privacy bound [49]. Interested readers can refer to Appendix A.1 for a brief introduction to private kNN and our implementations.

In experiments, we use the state-of-the-art private pseudo-labeling method, private kNN [56], to label the subsampled open-source data with the assistance from the labeled client dataset. To reduce the sensitivity of private kNN w.r.t. the threshold, we instead enforce the number of the selected labels to be 600 and balance the sizes by selecting the top-60 samples with the highest confidence per class. Then, we adopt the popular semi-supervised learning method, FixMatch [46], to train the classifier. In Table 4, we report the results when cloud features are distributionally biased from the client ones but they share the same class set. For each domain choice of client data, we will use the other domains as the cloud dataset. Distinct from prior studies, e.g., in [56] or [40], we assume 80-90% of the cloud data are out of the client distribution and are heterogeneously aggregated from different domains, casting greater challenges in learning. On the Digits, we eliminate all ID data from the cloud set to harden the task. Both on Digits and DomainNet datasets, our method consistently outperforms the two sampling baselines under the similar privacy costs. The variance of privacy costs is mainly resulted from the actual sampling sizes. Though simply adopting K-Center outperforms the random sampling, it still presents larger gaps compared to our method in multiple domains. For instance, in Quickdraw domain, given 3000 sampling budget, the K-Center method performs poorly and is even worse than a random sampling.

## 4.2 Qualitative Study

To better understand the proposed method, we conduct a series of qualitative studies on client labeling. We use two DomainNet datasets as a representative benchmark in the studies.

Table 5: Ablation study of the proposed method on DomainNet. Test accuracy of the client labeling case is reported.

| Proximity | Diversity | Clipart | Infograph | Painting | Quickdraw | Real | Sketch | Average |
|---|---|---|---|---|---|---|---|---|
| ✗ | ✗ | 30.7±2.2 | 21.6±4.9 | 44.0±3.1 | 39.0±5.6 | 59.9±3.6 | 47.2±3.4 | 40.4 |
| ✓ | ✗ | 25.5±5.7 | 21.1±0.5 | 41.4±5.3 | 31.3±1.3 | 60.3±2.3 | 31.9±2.9 | 35.2 |
| ✗ | ✓ | **39.3**±3.6 | 18.2±2.7 | 46.6±2.5 | 36.1±2.9 | **63.8**±1.9 | 47.7±2.8 | 42.0 |
| ✓ | ✓ | 33.2±5.9 | **23.8**±2.2 | **47.4**±3.3 | **39.8**±7.7 | 62.9±1.9 | **51.7**±1.2 | **43.2** |

1) **Ablation study**. In Table 5, we conduct ablation studies to evaluate the effect of different objectives introduced by ECOS, following the client labeling benchmark with a 3000 sample budget. Without proximity and diversity objectives, we let the baseline be the naive random sampling. We first include the proximity objective, where we greedily select samples from top-scored clusters until the budget is fulfilled. However, we find that the naive proximity objective results in a quite negative effect compared to the random baseline. The failure can be attributed to the nature of clustering, which includes more similar samples and thereby lacks diversity. When diversity is encouraged and combined with the proximal votes, we find the performance is improved significantly in multiple domains and on average. Now with diversity, the proximity objective can further improve the sampling in Painting, Quickdraw, and Sketch domains significantly. 2) **Visualize cluster selection**. In Fig. 3, we demonstrate that the CC can effectively reject OoD centroids. Also, it is interesting to observe that when multiple centroids are distributed closely, then they will compete with each other and reject the redundant ones consequently. 3) **Efficiency**. In Section 3, we studied the communication efficiency

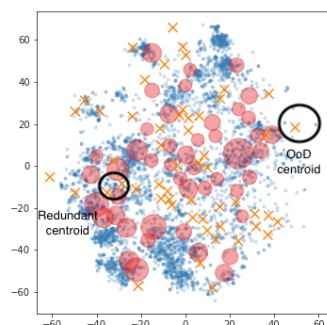

Figure 3: Demonstration of the centroids qualified by private CC. We use blue dots to represent the client data from Real domain of Domain-Net. For the data of the cloud domains, larger circles represent centroids with higher CC and orange crosses are rejected OoD centroids.

theoretically. Empirically, ECOS only needs to upload 100 bytes of the CC scores in all experiments, while traditional outsourcing needs to upload the dataset, which is about 198MB for the lowest load in DomainNet (50% of Sketch domain data for client). Even counting the download load, ECOS only needs to download 45MB of the pre-trained ResNet18 feature extractor together with 51KB data of centroid features, which is much less than data uploading. More detailed evaluation of the efficiency is placed in Appendix B.4 and we study how sample budgets and privacy budgets trade off the accuracy of ECOS models in Appendix B.3.

## 5  Conclusion

In this paper, we explore the possibility of outsourcing model training without access to the client data. To reconcile the data shortage from the target domain, we propose to find a surrogate dataset from the source agnostic public dataset. We find that the heterogeneity of the open-source data greatly compromises the performance of trained models. To tackle this practical challenge, we propose a collaborative sampling solution, ECOS, that can efficiently and effectively filter open-source samples and thus benefits follow-up learning tasks. We envision this work as a milestone for the private and efficient outsourcing from low-power and cost-effective end devices.

We also recognize open questions of the proposed solution for future studies. For example, the public dataset may require additional data processing, e.g., aligning and cropping for improved prediction accuracy. In our empirical studies, we only consider the computer vision tasks, though no assumption was made on the data structures. We expect the principles to be adapted to other data types with minimal efforts. More data types, including tabular and natural-language data, will be considered in the follow-up works. Take the language data for example: BERT is among the most popular pre-trained models for extracting features from sentences [16], upon which ECOS can assistant to sample proxy data from massive public dataset, for instance, WikiText-103 [35] based on 28,475 Wikipedia articles. Alternative to the $L_2$-norm based proximity objectives, advanced semantic features could also enhance the sampling effectiveness of ECOS in language data [54]. More discussions on the social impacts of this work are enclosed in Appendix A.5.

## Acknowledgments and Disclosure of Funding

This research was funded by Sony AI. This material is based in part upon work supported by the National Science Foundation under Grant IIS-2212174, IIS-1749940, Office of Naval Research N00014-20-1-2382, and National Institute on Aging (NIA) RF1AG072449.

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
