|---|---|---|---|---|---|---|---|---|---|---|---|---|
| Budget | Method | Acc (%) ↑ | $\epsilon$ ↓ | Acc (%) ↑ | $\epsilon$ ↓ | Acc (%) ↑ | $\epsilon$ ↓ | Acc (%) ↑ | $\epsilon$ ↓ | Acc (%) ↑ | $\epsilon$ ↓ | Acc (%) ↑ |
| | Ours | **84.2**±2.3 | 5.35 | 47.9±3.1 | 5.32 | **86.1**±1.0 | 5.35 | 68.6±1.6 | 5.35 | **58.4**±1.9 | 5.35 | **69.0** |
| 5000 | K-Center | 81.9±3.4 | 5.34 | 48.4±1.2 | 5.33 | 82.1±1.2 | 5.34 | 69.4±1.9 | 5.34 | 55.4±2.0 | 5.34 | 67.4 |
| | Random | 81.8±4.1 | 5.34 | 45.3±3.0 | 5.29 | 81.2±2.3 | 5.34 | 65.9±2.7 | 5.34 | 55.5±2.6 | 5.34 | 65.9 |

| Sampling | | Clipart | | Infograph | | Painting | | Quickdraw | | Real | | Sketch | | Average |
|---|---|---|---|---|---|---|---|---|---|---|---|---|---|---|
| Budget | Method | Acc (%) ↑ | $\epsilon$ ↓ | Acc (%) ↑ | $\epsilon$ ↓ | Acc (%) ↑ | $\epsilon$ ↓ | Acc (%) ↑ | $\epsilon$ ↓ | Acc (%) ↑ | $\epsilon$ ↓ | Acc (%) ↑ | $\epsilon$ ↓ |

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

# A  More Method Details

In this section, we elaborate on additional technical and theoretical details of our paper.

## A.1  Automated client labeling via private kNN

Here, we briefly introduce the main idea of client labeling by private kNN. Given $C$ classes, labeling is done by nearest neighbor voting:

$$f(x) = \arg\max_{c \in [C]} A_\epsilon(v_c), \ v_c = |\{(x', y') \in \mathbb{N}_K(x)|y' = c\}|,$$

where $\mathbb{N}_K(x)$ is a set including the $K$-nearest neighbors of $x$ in the client dataset. $A_\epsilon$ is a privatization mechanism complying with the notion of $\epsilon$-Differential Privacy (DP) [19]. In brief, privatization is done by adding Gaussian noise to a value with finite sensitivity. To filter out potential wrong labels, the client only returns high-confident samples by screening. Let the confidence of a pseudo label be $s(x) = \max_{c \in [C]} A_\epsilon(v_c)$ which is also privatized by the Gaussian noise mechanism. We find that the original version of screening may suffer from a large imbalance of pseudo labels. Per class, we screen the pseudo labels by selecting the top-60 confident samples given 600 labeling budget.

## A.2  Improving client labeling

Because of the noise mechanism for privacy protection, the client labeling may be quite random if the selected samples are hard to discriminate. Thus, we propose to improve the discrimination of selected samples in advance, when the ECOS selects samples for labeling. First, we estimate the discrimination by the confidence in the ECOS. The ECOS confidence is defined by the vote count of the highest-voted class subtracting the one of the second highest one, denoted as $v_r^{\mathrm{conf}}$. To merit the balancedness of samples, we filter the clusters to keep the top-70% samples with the highest confidence *per class*. When decompressing the clusters on the cloud, we incorporate the confidence into the sampling score by $v_r' = \psi\left[(v_r^{\mathrm{conf}} + v_r)/2\right]$ where $v_r$ is the original score.

## A.3  Privacy Accountant for ECOS

To understand the privacy cost of ECOS, we review the techniques that are essential to establish the privacy bound.

**Definition A.1** (Differential Privacy [19])**.** Suppose $\epsilon$ and $\delta$ are two positive constants. A randomized algorithm $\mathcal{M} : \mathcal{X} \to \mathcal{Y}$ is $(\epsilon, \delta)$-DP if for every pair of neighboring datasets $X, X' \in \mathcal{X}$, and every possible measurable output set $Y \subset \mathcal{Y}$ the following inequality holds:

$$\Pr[\mathcal{M}(X) \in Y] \le e^\epsilon \Pr[\mathcal{M}(X') \in Y] + \delta,$$

where $\Pr[\cdot]$ denotes the probability of a given event.

DP provides a way to quantify the privacy risk (termed as the difference between two probability given a pair of similar but different inputs) in the probability of $\delta$. Though DP is a simple notion for risks, the estimation of a tight privacy bound is still challenging. For this reason, RDP is proposed an alternative tool.

**Definition A.2.** (Rényi Differential Privacy (RDP) [36]) A randomized algorithm $\mathcal{M} : \mathcal{X} \to \mathcal{Y}$ is $(\alpha, \epsilon)$-RDP with order $\alpha > 1$ if for all neighboring datasets $X, X'$ the following holds

$$D_\alpha(\mathcal{M}(X)||\mathcal{M}(X')) \le \epsilon,$$

where $D_\alpha(\cdot||\cdot)$ is the Rényi divergence between two distributions.

The RDP and DP can be connected by Lemma A.1.

**Lemma A.1.** *If a mechanism $\mathcal{M}$ satisfies $(\alpha, \epsilon)$-RDP, then it also satisfies $(\epsilon + \frac{\log 1/\delta}{\alpha - 1}, \delta)$-DP.*

To reveal the potential privacy risks, we give a theoretical bound on the privacy cost based on DP in Lemma A.2. The proof of Lemma A.2 is similar to Theorem 8 in [56] without confidence screening.

**Lemma A.2.** *Suppose the subsampling rate $\gamma$ and noise magnitude $\sigma$ of the ECOS are positive values such that $\gamma \leq \min\left\{0.1, \sigma\sqrt{\log(1/\delta)/6}\right\}$ and $\sigma \geq 2\sqrt{5}$. The total privacy bound of the ECOS scoring $m = |\hat{D}^q|$ query samples with $n = |\hat{D}^p|$ private client samples is $(\epsilon, \delta)$-DP with $\delta > 0$ and*

$$\epsilon = \mathcal{O}(\frac{\gamma}{\sigma}\sqrt{\log(1/\delta)}), \tag{4}$$

*if $v_r$ in Algorithm 1 is estimated by using $\lceil \gamma n \rceil$ samples randomly subsampled from $\hat{D}^p$ with replacement and is processed by $\tilde{v}_r = v_r + \mathcal{N}(0, \sigma^2 I)$.*

*Proof.* When one sample is absent from the private client dataset, the scores $[v_1, \ldots, v_R]$ will differ by 2 if without subsampling. By Lemma 11 of [56], the subsampled Gaussian mechanism accounted by the RDP is

$$\epsilon(\alpha) \leq \frac{24\gamma^2\alpha}{\sigma^2}$$

for all $0 < \alpha \leq \frac{\sigma^2 \log(1/\gamma)}{2}$, $\gamma \leq 0.1$ and $\sigma \geq 2\sqrt{5}$. By Lemma A.1, we can convert the RDP inequality to the standard $(\epsilon, \delta)$-DP as

$$\epsilon = \frac{24\gamma^2\alpha}{\sigma^2} + \frac{\log(1/\delta)}{\alpha - 1}.$$

Let $\alpha$ be $1 + \frac{\sqrt{\log(1/\delta)}}{\sqrt{\frac{24\gamma^2}{\sigma^2}}}$. Thus,

$$\epsilon = \frac{24\gamma^2}{\sigma^2} + 4\frac{\gamma}{\sigma}\sqrt{6\log(1/\delta)} \leq 8\frac{\gamma}{\sigma}\sqrt{6\log(1/\delta)},$$

where the last inequality is derived by the given range of $\gamma$. This thus completes the proof. $\qquad\square$

The above bound implies that the privacy cost for our method is invariant w.r.t. the scale of the query dataset $\hat{D}^q$, and only depends on the DP parameters. Note that the Lemma A.2 is an asymptotic bound which requires some strict conditions on $\gamma$ or other variables. In practice, we leverage the tool of analytic privacy accountant through a numerical method [49], with which we can relax the strict conditions.

## A.4 Theoretical Analysis

Though empirical results show that more accurate models can be trained on ECOS-sampled datasets, it remains unclear how the cloud dataset and the training process affect the model performance on the client. In this section, we provide the proof of Theorem 3.1 using the domain generalization bound as stated in Theorem A.1.

**Theorem A.1** (Domain-adaptation learning bound from [7]). *Suppose two domains with distributions $\mathcal{D}_s$ and $\mathcal{D}_t$. Let $\mathcal{H}$ be a hypothesis space of $VC$-dimension $d$ and $D_s$ be the dataset induced by samples of size $N$ drawn from $\mathcal{D}_s$, respectively. Then with probability at least $1 - p$ over the choice of samples, for each $f \in \mathcal{H}$,*

$$L(f, \mathcal{D}_t) \leq L(f, D_s) + \frac{1}{2}d_{\mathcal{H}\Delta\mathcal{H}}(\mathcal{D}_s, \mathcal{D}_t) + 4\sqrt{\frac{2d\log(2N) + \log(4/p)}{N}} + \xi, \tag{5}$$

*where $\xi$ is the optimal loss, i.e., $\min_f L(f, \mathcal{D}_s) + L(f, \mathcal{D}_t)$, and $d_{\mathcal{H}\Delta\mathcal{H}}(\mathcal{D}_s, \mathcal{D}_t)$ denotes the divergence between domain $s$ and $t$.*

*Proof of Theorem 3.1.* Our proof is mainly based on Theorem A.1. By definition, $S$ contains data sampled from the mixture of distributions, $\sum_{r=1}^{R}\mathcal{D}_r^q$. Apply Theorem A.1 to attain

$$L(f_\theta, \mathcal{D}^p) \leq L(f_\theta, S) + \frac{1}{2}d_{\mathcal{H}\Delta\mathcal{H}}(\sum_{r=1}^{R}\mathcal{D}_r^q, \mathcal{D}^p) + 4\sqrt{\frac{2d\log(2|S|) + \log(4/p)}{|S|}} + \xi. \tag{6}$$

And we have

$$d_{\mathcal{H}\Delta\mathcal{H}}\left(\sum_{r=1}^{R}\mathcal{D}_r^q, \mathcal{D}_t\right) \leq \sum_{r=1}^{R}\alpha_r d_{\mathcal{H}\Delta\mathcal{H}}(\mathcal{S}_r, \mathcal{D}_t), \tag{7}$$

which was proved in [42] (proof of Theorem 2). Plugging Eq. (7) into Eq. (6), we can get the conclusion. □

Now, we consider $f_\theta$ to be trained by the widely-adopted gradient descent method. We first present Theorem A.2 which characterizes the empirical error bound after $T$ iterations. Substitute Eq. (8) into Eq. (3). Then we can obtain Lemma A.3.

**Theorem A.2** (Convergence bound, e.g., from [8])**.** *Suppose the model $f$ is parameterized by $\theta$ initialized as $\theta_0$. Let the loss function $L(f_\theta, D)$ be $\mu$-strongly convex and $G$-smooth w.r.t. $\theta$, and assume that the global minima $\theta^*$ exists. Then after $T$ iterations, gradient descent with a fixed learning rate $\eta \leq 1/G$ satisfies*

$$L(f_{\theta_T}, D) \leq L^* + (1 - \mu/G)^T \ell_0(\theta_0, D) \tag{8}$$

*where $L^* = \min_\theta L(f_\theta, D)$ and $\ell_0(\theta_0, D)$ is the initial error gap, i.e., $|L(f_{\theta_0}, D) - L^*|$.*

**Lemma A.3.** *Suppose assumptions in Theorems 3.1 and A.2 holds. Let $L(\cdot, \cdot)$ be a loss function on a hypothesis and a dataset (for empirical error) or a distribution (for generalization error). If $f$ is governed by the parameter $\theta$ trained on $S$ via $T$-iteration gradient descent and belongs to a hypothesis space $\mathcal{H}$ of $VC$-dimension $d$, then with probability at least $1 - p$ over the choice of samples, the inequality holds,*

$$L(f_{\theta_T}, \mathcal{D}^p) \leq (1 - \frac{\mu}{G})^T \ell_0(\theta_0, S) + \frac{1}{2}\sum_{r=1}^{R}\alpha_r d_{\mathcal{H}\Delta\mathcal{H}}(\mathcal{D}_r^q, \mathcal{D}^p)$$

$$+ 4\sqrt{\frac{2d\log(2|S|) + \log(4/p)}{|S|}} + \xi', \tag{9}$$

*where $\xi' = \min_\theta \{L(f_\theta, S) + L(f_\theta, D^p)\} + \min_\theta L(f_\theta, S)$, and $d_{\mathcal{H}\Delta\mathcal{H}}(\mathcal{D}, \mathcal{D}')$ denotes the distribution divergence.*

## A.5  Social Impacts

The conflict between the concerns on data privacy and demands for intensive computation resources for machine learning has composed the main challenge in training outsourcing. In this work, we devote our efforts to outsourcing with uploading data by leveraging authorized or public datasets. As the public datasets commonly available in many applications are collected from multiple data sources and thus tend to be non-identically distributed as the client data, it casts new challenges to use the public in place of the client dataset. Our method addresses this problem with accountable privacy cost and low communication and computation complexity. Therefore, the proposed ECOS provides a promising solution to mitigate the aforementioned conflict between the privacy and computation desiderata. Therefore, users from a broader spectrum can benefit from such a method to confidentially conduct cloud training.

## A.6  Connection to Federated Learning

Both our method and federated learning (FL) [34] consider protecting data privacy via not sharing data with the cloud. The key difference between FL and our concerned problem (training outsourcing) is that FL requires clients to conduct training while ECOS outsources the training to the cloud server. Since ECOS does not require local training, it can be ad-hocly plugged into FL to obtain an auxiliary open-source dataset for enhancing the federated training. ECOS can be used either before federated training (when a pre-trained model is required) or during federated training (when the pre-trained model can be replaced by the on-training model).

# B  Experimental Details and More Experiments

Complementary to the main content, we provide the details of the experiment configurations to merit the reproducibility. We also conduct more qualitative experiments to understand the efficiency and effectiveness of the proposed method.

### B.1 Experimental Details

We organize the case-specified configurations into three cases and discuss the general setups first.

For Digits, we train the model for 150 epochs. We adopt a convolutional neural network for Digits in Table 6 and ResNet18 for DomainNet. To solve the learning problems including FixMatch and distillation-based compression, we use stochastic gradient descent with the momentum of 0.9 and the weight decay of $5 \times 10^{-4}$. We use $s = 5$ for the scale function $\psi_s$ on DomainNet and $s = 1$ on Digits. When not specified, we noise the ECOS query with the magnitude as 25. In Case 1 and 2, we reduce the noise magnitude to 10 for DomainNet, since the two queries can bear more privacy costs to trade for higher accuracy.

**Case 1: Selective manual labeling**. We make use of the off-the-shelf ResNet18 is pre-trained on the ImageNet, which is widely accessible online. We adopt FixMatch for semi-supervised learning with the coefficient of 0.1 on the pseudo-labeled loss, the moving average factor of 0.9, and the batch size of 64 for DomainNet and 128 for Digits. To avoid feature distorting, we warm up the fine-tuning by freezing all layers except the last linear layer with a learning rate of 0.01. After 30 epochs, we fine-tune the model end to end until 80 epochs to avoid overfitting biased data distributions.

**Case 3: Adaptive model compression**. We first pre-train a ResNet50 using all labeled open-source data for 100 epochs with a cosine-annealed learning rate from 0.1. The same warm-up strategy as Case 1 is used here. To extract the knowledge from ResNet50, we combine the knowledge-distillation (KD) loss $L_{KD}$ and cross-entropy loss $L_{CE}$ by $0.1 \times L_{KD} + 0.9 \times L_{CE}$ and calculate the losses on the selected samples only. The temperature in the KD loss is set to be 10.

**Case 2: Automated client labeling**. For the cloud training, we adopt the same configuration as the selective manual labeling. For private kNN, we let the client release 600 labels with class-wise confidence thresholds described in the last section. We noise the labeling in the magnitude of 25 and the confidence in the magnitude of 75. For both datasets, we subsample 80% client data per labeling query to reduce the privacy cost.

Table 6: The structure of the conventional neural network for the Digits dataset.

| Layer name | PyTorch pseudo code |
|---|---|
| conv1 | Conv2d(1, 64, kernel_size=(5, 5), stride=(1, 1)) |
| bn1 | BatchNorm2d(64, eps=1e-05, momentum=0.1) |
| conv1_drop | Dropout2d(p=0.5, inplace=False) |
| conv2 | Conv2d(64, 128, kernel_size=(5, 5), stride=(1, 1)) |
| bn2 | BatchNorm2d(128, eps=1e-05, momentum=0.1) |
| conv2_drop | Dropout2d(p=0.5, inplace=False) |
| fc1 | Linear(in_features=2048, out_features=384, bias=True) |
| fc2 | Linear(in_features=384, out_features=192, bias=True) |
| fc3 | Linear(in_features=192, out_features=11, bias=True) |

We conduct our experiments on the Amazon Web Service platform with 4 Tesla T4 GPUs with 16GB memory and a 48-thread Intel CPU. All the code is implemented with `PyTorch` 1.11. To account for the privacy cost, we utilize the open-sourced `autodp` package following the private kNN.

### B.2 Effect of Parameters

To better understand the proposed method, we study the effect of the important hyper-parameters. To this end, we consider the selective labeling task with Digits, keeping 50% of the SVHN dataset at the client end. Both the ID+OoD and OoD cases are evaluated to reveal the method's effectiveness under circumstances with various hardness. Also, we study how the score scale $s$ affects the ID ratios (denoted as the ID TPR) in the selected set and the number of effective samples. We only examine the ID TPR corresponding to the proximity objective in Eq. (2) if known ID samples are present on the cloud, namely in the *ID+OoD* case. In the middle panes of Figs. 4 and 5, we show that our method can effectively improve the ID TPR against the inherent ratio of ID samples on the cloud.

**Effect of the compression size $R$**. In Fig. 4, we evaluate $R$ in terms of the test accuracy. When the budget is small (1k and 2k budgets in the ID+OoD case), it is essential for the cloud server to sense the client distribution with higher accuracy via more queries. Therefore, a larger $R$ is desired, which can increase the portion of ID samples in the selected set, as shown in the middle pane of Fig. 4. Considering that a higher burden on communication comes with a larger $R$, the value of 100 leads to a fair trade-off to the accuracy in which case the ID TPR reaches a peak.

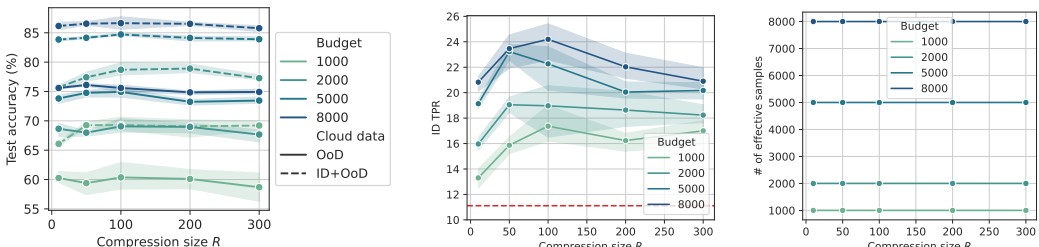

Figure 4: Vary the compression size $R$ and evaluate the test accuracy, ID ratios (%) in selected samples and the number of effectively selected samples. The red horizontal line indicates the ratio of ID samples in the whole cloud dataset.

Given a larger budget, e.g., 8000, increasing $R$ may lower the ID TPR. We attribute the decline to the limited size of the client dataset and privacy constraints. Given more clusters (i.e., $R$), the expected number of votes (proportional to the score) for each cluster will be reduced and is badly blurred by the DP noise. Thus, the ID TPR will decrease simultaneously, regardless of which the test accuracy is not significantly affected.

For the OoD case which is relatively harder for sampling due to the lack of true ID data, the parameter sensitivity is weakened, though the compression size of $100$ is still a fair choice, for example, bringing in $1 - 2\%$ gains in the 5k, 8k cases comparing the worst cases.

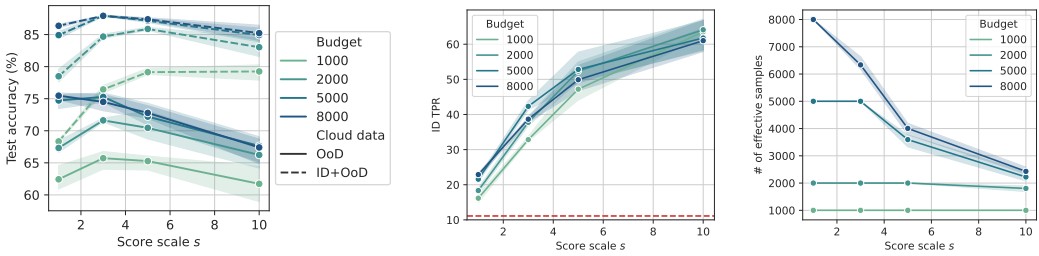

Figure 5: Vary the score scale $s$ in terms of test accuracy, ID ratios (%) in selected samples and the number of effectively selected samples (which could be smaller than the budget). The red horizontal line indicates the ratio of ID samples in the whole cloud dataset.

**Effect of the score scale** $s$. The score scale $s$ decides the sensitivity of the sampling in the sense of proximity. A larger $s$ means that the ECOS will prioritize the proximity more during sampling. In Fig. 5, we present the ablation study of $s$. A larger $s$ is preferred when the budget becomes limited because it increases the ID TPR effectively. Though not significantly, an overly large $s$ has a significantly negative influence on the accuracy, especially for the OoD case. The reason for the negative impact of $s$ on a large budget can be understood by probing the number of effective samples. For budgets larger than 2000, the effectively selected samples are reduced with heavily scaled scores (e.g., $s \geq 3$) where the ECOS will concentrate its selection into very few clusters and eliminate the rest clusters strictly.

### B.3 Evaluation of Sample and Privacy Efficiency

**Effects of sample budgets.** In Fig. 6, we compare the sample efficiency in the selective labeling task with Digits, keeping $50\%$ of the SVHN dataset at the client end. We obtain the upper-bound accuracy in the ideal case via random sampling when the cloud dataset distributes identically (ID) as the client dataset. When OoD data are included in the open-source cloud dataset (ID+OoD), the training becomes more demanding for the labeled samples. If none of the iid samples presents in the cloud set (OoD), the accuracy decreases quickly with the same labeled samples. In comparison, informative sampling by K-Center slightly improves the accuracy by different budgets and the proposed ECOS significantly promotes the sample efficiency. With ECOS, $8 \times 10^3$ labeled samples in the ID+OoD case achieves comparable accuracy as the ideal case, while baselines remain large gaps. Both in

ID+OoD and OoD cases, our method yields competitive accuracy (at the $4 \times 10^3$ budget) versus the best baseline results using only half of the labeled data (at the $1 \times 10^4$ budget), dramatically cutting down the cost for manual labeling.

On observing the gains in sample efficiency, readers may also notice that our method induces additional costs at privacy, as compared to the baselines. We point out that the cost is constant w.r.t. the sampling budget and is as neglectable as $(0.22, 10^{-5})$-DP. It is worth noticing that the cost is independent of the hyper-parameters of the ECOS because the ECOS communication is a *single* query for each private sample (so as for the private dataset), even if we increase the size of the query set (i.e., the compression size $R$). In practice, the client can control the privacy risk (namely, the privacy cost) flexibly by adjusting the noise magnitude and the subsampling rate.

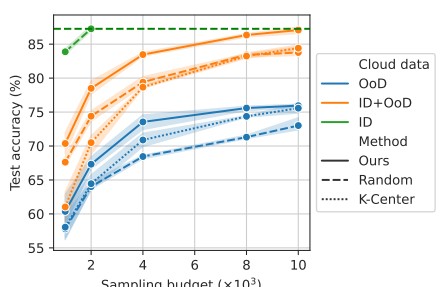

Figure 6: Evaluation of the sample efficiency on selective labeling. The green horizontal line implies the ideal case when all ID cloud data are labeled.

**Privacy-accuracy trade-off.** Additionally, we evaluate how the privacy costs affect the In Fig. 7, we study the relationship between privacy cost and performance by varying the noise scale $\sigma$ of the ECOS in label outsourcing. The experiment is conducted on SVHN client data for label outsourcing. Interestingly, the sampling effectiveness is not very sensitive to the noise scale. A very low privacy cost $(0.08)$ can be achieved with noise as large as 70 and accuracy as high as $83.7\%$. The success could be attributed to the low dimension of queries (100 clusters) to the private dataset, resulting in the efficiency of privacy costs.

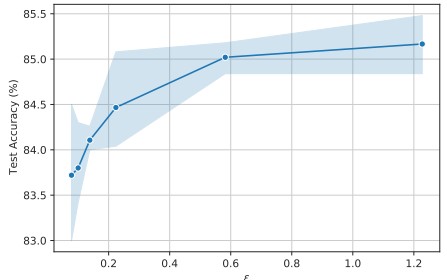

Figure 7: Evaluation on how the privacy costs affect the performance.

## B.4 Evaluation of Communication and Computation Efficiency

When improving the sample efficiency, we also need to take care of the communication and computation overheads brought by the ECOS. We examine the two kinds of efficiency by the same experiment configurations as in the last section.

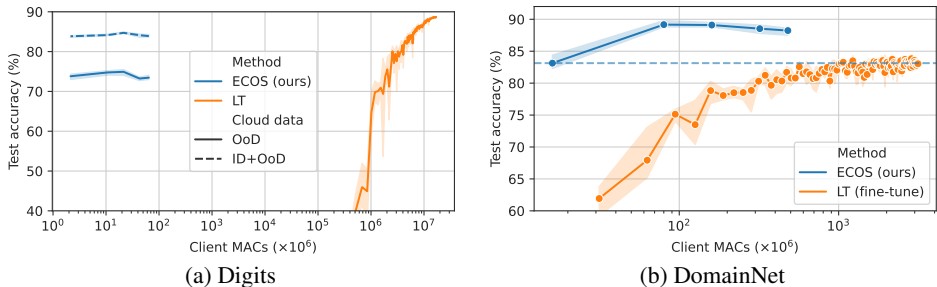

(a) Digits

(b) DomainNet

Figure 8: With the 5000 budget, we evaluate the computation efficiency. The efficiency of locally-training (LT) on the DomainNet is enhanced by linear fine-tuning where only the linear classifier head is locally trained.

**Computation efficiency.** In Fig. 8, we compare the computation efficiency of our method to the local training (LT). We utilize the multiplication-and-addition counts (MACs) as the metric of computation (time) complexity, which is hardware-agnostic and therefore is preferred here. For a fair comparison, we tune the learning rate in $\{0.1, 0.01, 0.001\}$ with the cosine annealing during training and the number of epochs in $\{20, 50, 100\}$ of the LT to achieve a fair trade-off between the computation cost and test accuracy. For ECOS, since the computation cost linearly increases by the compression size (as shown in Fig. 9), we vary the compression size to check the performance when increasing computation costs. On Digits, we observe a large computation save by our method, even if the cost of our method will gradually increase by the size $R$ of the compressed query set.

Similar experiments are also run on the large-sized images using the DomainNet dataset (ID+OoD case), where the cost for extracting features is steeply increased by using a deep network (ResNet18). Recently, the most popular strategy for cloud training is two-phase learning: pre-training a model on the cloud using ImageNet and fine-tuning the linear classifier head on the client. Considering the large cost of feature extraction, we only let the client pre-extract features once only. Thus, the local training is as efficient as training a *linear* layer on extracted features. In Fig. 8, our method outperforms the local training a lot using much fewer MACs for data matching. Because all training is outsourced to the cloud, our method enables the end-to-end fine-tuning of the model resulting in better test

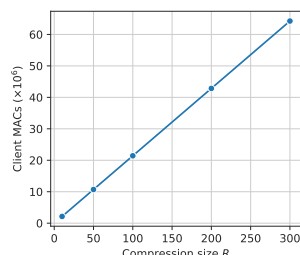

Figure 9: The linearly growing computation cost by increasing the compression size on the Digits dataset.

accuracy. Even if the LT trains longer with higher computation costs, the test accuracy of the ECOS with the least MACs is comparable to the best performance of LT at around $10^9$ MACs, where the ECOS only utilizes the 10% of MACs by LT.

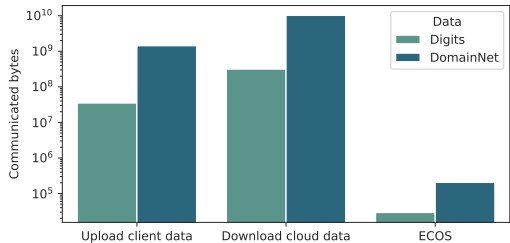

Figure 10: Evaluate the communication efficiency.

**Communication efficiency.** We also compare the communication efficiency to the full cloud training (via uploading the whole client dataset) and fully client training (via downloading cloud dataset) in Fig. 10. For the ECOS, we let the size of the query set be 100, which is the default configuration in our experiments. Because the ECOS only communicates a few low-dimensional features (for example, 512-dimensional ResNet-extracted features for DomainNet and 72-dimensional HOG features for Digits), it costs much fewer bytes compared to traditional outsourcing by uploading the client data. To be concrete, we also present the cost of downloading the cloud data and it is way more expensive than the rest two methods.