# OpenReview forum: "Outsourcing Training without Uploading Data via Efficient Collaborative Open-Source Sampling"
_NeurIPS.cc/2022/Conference — NeurIPS 2022 Accept_

### Official Review · Reviewer_tw1P · 2022-07-07

**Rating:** 6
**Confidence:** 4
**Soundness:** 3 good
**Presentation:** 3 good
**Contribution:** 3 good

**Summary:**

The authors considered utilizing public open-source data rather than sensitive private data when outsourcing the computation of machine learning under strong privacy constraints. To tackle the problem that the heterogenous public data from various domains, which may be not IID with the target-domain data and may degrade the model performance, the authors proposed the ECOS framework, which filters the public data according to the local private data.

**Questions:**

1. On the one hand, more data benefit training from reducing overfitting risk. On the other hand, out-of-distribution (OoD) samples degrade the model performance. The authors consider filtering public data to get rid of OoD samples, which, however, may reduce the amount of training data. Does the proposed ECOS framework strike a balance between more training data and high-quality training data, theoretically and empirically?

2. Federated learning considers a similar scenario of privacy constraints. We recommend the authors to compare their setting with federated learning in the related work.


**Limitations:**

See my comments on more theoretical analysis and comparison with federated learning.

**Strengths And Weaknesses:**

### Strengths
**Originality:** The authors considered a new problem of outsourcing machine learning under privacy constraints. This proposed framework clusters the public data and distributes the cluster centroids to the clients. The clients then use the private local data to score the centroids and then use the scores to filter the heterogeneous public data. This new framework works intuitively to alleviate the problem of data heterogeneity.

**Quality:** Compared with existing baselines, the proposed ECOS framework gains accuracy improvement in the evaluation. The authors also provided visualization which illustrate the effects of the ECOS framework.

**Clarity:** The paper is well organized and the proposed method is clear.

### Weaknesses

**Quality:** The theoretical analysis is not very sufficient, e.g., about how much accuracy the data filtering process can improve (compared with using all public data).

**Significance:** Federated learning considers the similar problem: users cannot upload raw data due to privacy concerns. There has been many research work in this field, which may also help address the problem considered by the authors.

---

> ### Author Response · Authors · 2022-08-02
> **Responses to Reviewer tw1P**
>
> We appreciate that you point out the originality, quality, and clarity of our paper. In the following responses, we tried our best to answer your quite insightful comments.
>
> Questions:
> > **Q1**: On the one hand, more data benefit training from reducing overfitting risk. On the other hand, out-of-distribution (OoD) samples degrade the model performance. The authors consider filtering public data to get rid of OoD samples, which, however, may reduce the amount of training data. Does the proposed ECOS framework strike a balance between more training data and high-quality training data, theoretically and empirically?
>
> We sincerely appreciate the insightful question.
> 1) We want to reiterate that we consider the number of samples as a constraint due to limited communication and computation bandwidth in practice, rather than a changeable parameter. Within a constant budget, our method can outperform baselines in terms of model accuracy (representing data quality), which is demonstrated in Tables 2-4. A detailed evaluation of the effect of the sample budget is provided in Fig 6.
> 2) To balance data quality and data size quantitatively, we provided a tunable parameter, score scale s (introduced at the end of the 2nd paragraph of page 5), which controlled the filtering sensitivity by reshaping the score function with the non-linear function $\psi$.
> In Figure 5 of Appendix B.2 (which has been included in our initial submission), we empirically show that given a large enough budget (e.g., >2000) and a ID + OoD open-source dataset, a larger s would result in higher in-distribution true positive rate (ID TPR) which means more in-distribution samples or higher data quality, but meanwhile eliminated more samples from the budget which means smaller data sizes.
> 3) In addition, we empirically studied the model accuracy caused by the above trade-off in Figure 5. When the sampling budget is sufficient, e.g., in an 8000 budget and ID+OoD case, a moderate value of s, e.g., 3, can maximize the accuracy. The reason is that the large budget allows the sampling to include most in-distribution samples and including more out-of-distribution samples at the time will help the generalization. For smaller budgets, a larger s works better, where the proximity is more important for the model transfer to the target dataset.
> 4) We would like to conduct more rigorous theoretical studies in our future work. One possible way is to analyze the trade-off of the data quality and sizes on the expected model losses, based on the domain adaption theorems. One insight based on [A] is that the model performance on the client dataset (target domain) should be proportional to the proximity governed by $s$ (corresponding to the domain divergence) and inversely proportional to the samples size of the filtered cloud dataset (corresponding to the data size of the source domain).
>
> [A] Ben-David, S., Blitzer, J., Crammer, K., Kulesza, A., Pereira, F., & Vaughan, J. W. (2010). A theory of learning from different domains. Machine learning, 79(1), 151-175.
>
> > **Q2**: Federated learning considers a similar scenario of privacy constraints. We recommend the authors to compare their setting with federated learning in the related work.
>
> We updated our paper with updated related work on Page 2 and more discussion on the connection to federated learning in Appendix A.5. In brief, the key difference between FL and our concerned problem (training outsourcing) is that FL requires clients to do training, while ECOS outsources the training process to the cloud server. Since ECOS does not require local training, ECOS can be ad-hocly plugged into FL to obtain an auxiliary open-source dataset on the cloud. The auxiliary dataset can be used for training on the cloud to mitigate the overfitting potentially caused by the limited data sizes in clients. In addition, the ECOS is quite flexible which can be used either before federated training (when a pre-trained model is required) or during federated training (when the pre-trained model can be replaced by the on-training model).

---

> ### Author Response · Authors · 2022-08-07
> **Thanks to Reviewer tw1P**
>
> Thanks again for the valuable comments. Especially, we appreciate your positive comments on the originality, quality, and clarity of our work.
>
> We have now clarified the connection between our method and federated learning and show the empirical and potential theoretical analysis of the trade-off between data quality and sizes. Note that more detailed information is shown in our rebuttal summary.
>
> Please kindly let us know if other concerns remain. We are truly thankful for this opportunity to enhance our work and shall be most grateful for your feedback.

---

### Official Review · Reviewer_YJH7 · 2022-07-08

**Rating:** 6
**Confidence:** 3
**Soundness:** 3 good
**Presentation:** 4 excellent
**Contribution:** 3 good

**Summary:**

This paper proposes a method to efficiently sample from open-source data in order to train models that are then transferable to private local datasets. This enables the use of machine-learning-as-a-service (MLaaS) to train models on the cloud without the need to upload confidential data to a remote server. The authors develop a new sampling method, by scoring centroids of clustered features from open-source data samples, which then guides the selection of a proper subset of the open-source dataset for model training. The uploaded scores can further be protected by adding Gaussian noise, in order to account for a desired privacy cost in the context of differential privacy (DP). This then leads to a technique with minimal computational/bandwidth requirements for the client. The proposed method is evaluated for digit and object recognition tasks, trained using pseudo, manual or pre-trained supervisions, and is shown to outperform baseline methods on average.

**Questions:**

- Do the authors have any insight as to why the random baseline outperforms the authors’ method in some cases? This seems counterintuitive especially since this baseline is perfectly privacy-preserving, and by using random sampling one is polluting the training dataset with out-of-distribution samples.
- Does the K-center baseline require uploading the client dataset (or a transformed version of it) to the server? If so, this should be clarified since Tables 2-4 suggest that it is perfectly privacy-preserving.
- For DomainNet, roughly what portion of the selected proximal data corresponds to the same domain as the client dataset? This could show the effectiveness of the sampling method in picking in-distribution samples.


**Limitations:**

The main limitation of this method is that it requires the utilized public dataset to contain samples that are similar in distribution to the private dataset. The authors have adequately discussed this in the paper.

**Strengths And Weaknesses:**

This paper addresses an important problem, especially with the growing popularity of MLaaS. The proposed method is sound and well-described. I also appreciate the analysis of the algorithm with regard to different privacy, which allows a user to fine-tune the method according to their privacy cost. In most cases, the performance of the proposed method (in Tables 2-4) is within the confidence interval of another baseline method, and therefore the performance gains are not necessarily statistically significant. Nevertheless, configurable privacy and the computational/bandwidth efficiency of this method are solid contributions.

I think the evaluation can be improved in the following regards:
- While the baselines (K-center and random) provide a useful comparison between the authors’ method and more naive approaches, it would also help to include accuracies when a model is trained on the client dataset itself. This would show how much accuracy one would lose (if any) by using a privacy-preserving proxy dataset.
- I also think a sensitivity analysis with respect to the privacy cost (\epsilon) could be very informative. How does changing \epsilon affect the achieved accuracy?

---

> ### Author Response · Authors · 2022-08-02
> **[1/2] Responses on the suggestions given by Reviewer YJH7**
>
> We appreciate that your acknowledgement on the importance, rigor, and novelty of our paper. We also appreciate the insightful comments and suggestions. We hope our responses and new results can address your concerns.
>
> > Suggestions
> >
> > **S1**: While the baselines (K-center and random) provide a useful comparison between the authors’ method and more naive approaches, it would also help to include accuracies when a model is trained on the client dataset itself. This would show how much accuracy one would lose (if any) by using a privacy-preserving proxy dataset.
>
> We add the locally trained results at the end of Table 2 when the results should be the same in other cases. For convenience, we include these results below, where standard deviations are included in brackets. We observe that in DomainNet, the local training performs worse than the cloud training, but in Digits, the conclusion is reversed. The reason for the difference is that Digits a small-size image dataset (28x28) where a small local dataset is enough for training, while DomainNet requires more samples for training given the 244x244 image size.
>
>
> * DomainNet
>
> | budget | method | Clipart | Infograph | Painting | Quickdraw | Real | Sketch |
> | :-------- | :-------- | :-------- | :-------- | :-------- | :-------- | :-------- | :-------- |
> | 1000  | Ours | **88.4** 1.5 |  0.58 | **52.6** 0.9 |  0.58 | **90.4** 1.7 |  0.58 | **84.3** 1.6 |  0.58 | 92.1 1.2 |  0.58 | **87.2** 0.5 |  0.58 | **82.5** |
> | 1000  | K-Center | 86.8 0.3 |  0.00 | 50.5 0.9 |  0.00 | 89.1 1.4 |  0.00 | 27.2 1.8 |  0.00 | **92.5** 0.1 |  0.00 | 85.6 1.4 |  0.00 | 72.0 |
> | 1000  | Random | 86.9 0.8 |  0.00 | 47.4 2.7 |  0.00 | 88.6 0.1 |  0.00 | 77.9 2.4 |  0.00 | 91.4 0.3 |  0.00 | 86.9 0.5 |  0.00 | 79.9 |
> | 3000  | Ours | 93.2 0.4 |  0.58 | **58.1** 0.6 |  0.58 | 92.5 1.1 |  0.58 | **89.2** 0.9 |  0.58 | **94.4** 0.2 |  0.58 | 92.8 0.2 |  0.58 | **86.7** |
> | 3000  | K-Center | **93.5** 1.1 |  0.00 | 56.3 0.3 |  0.00 | **92.9** 0.3 |  0.00 | 60.5 8.7 |  0.00 | 94.1 0.2 |  0.00 | 92.1 0.7 |  0.00 | 81.6 |
> | 3000  | Random | 92.5 0.6 |  0.00 | 53.6 1.4 |  0.00 | 91.7 0.8 |  0.00 | 86.1 0.4 |  0.00 | 93.5 0.3 |  0.00 | **93.0** 0.2 |  0.00 | 85.1 |
> | -  | Local | 87.0 0.3 |  0.00 | 51.7 0.7 |  0.00 | 85.9 0.4 |  0.00 | 83.5 0.3 |  0.00 | 93.5 0.1 |  0.00 | 81.7 0.6 |  0.00 | **80.6** |
>
> * Digits
>
> | budget | method | MNIST | SVHN | USPS | SynthDigits | MNIST-M |
> | :-------- | :-------- | :-------- | :-------- | :-------- | :-------- | :-------- |
> | 2000  | Ours | **97.3** 0.1 |  0.22 | **68.7** 0.3 |  0.22 | 90.8 0.1 |  0.22 | **84.4** 0.6 |  0.22 | 70.4 0.6 |  0.22 | **82.3** |
> | 2000  | K-Center | 96.7 0.3 |  0.00 | 65.1 1.3 |  0.00 | 90.1 0.7 |  0.00 | 80.2 1.1 |  0.00 | 70.1 0.3 |  0.00 | 80.4 |
> | 2000  | Random | 96.5 0.3 |  0.00 | 64.0 0.3 |  0.00 | **91.6** 1.0 |  0.00 | 83.8 0.3 |  0.00 | **70.9** 0.6 |  0.00 | 81.4 |
> | 5000  | Ours | **98.1** 0.2 |  0.22 | **74.6** 1.0 |  0.22 | **93.5** 0.3 |  0.22 | **91.2** 0.4 |  0.22 | **74.5** 0.5 |  0.22 | **86.4** |
> | 5000  | K-Center | 97.9 0.2 |  0.00 | 72.3 0.6 |  0.00 | 92.7 0.9 |  0.00 | 89.6 0.3 |  0.00 | 74.0 0.5 |  0.00 | 85.3 |
> | 5000  | Random | 97.6 0.3 |  0.00 | 70.0 0.3 |  0.00 | 93.0 0.6 |  0.00 | 89.7 0.4 |  0.00 | 73.9 0.7 |  0.00 | 84.8 |
> | -  | Local | 99.1 0.1 |  0.00 | 88.8 0.2 |  0.00 | 98.9 0.1 |  0.00 | 96.4 0.2 |  0.00 | 88.8 0.2 |  0.00 | **94.4** |
>
>
> > **S2**:  I also think a sensitivity analysis with respect to the privacy cost (\epsilon) could be very informative. How does changing \epsilon affect the achieved accuracy?
>
> We add new results on the relationship between privacy cost and performance by varying the noise scale $\sigma$ of the ECOS in label outsourcing. The experiment is conducted on SVHN client data for label outsourcing. Interestingly, the sampling effectiveness is not very sensitive to the noise scale. A very low privacy cost (0.08) can be achieved with noise as large as 70 and accuracy as high as 83.7%. The success could be attributed to the low dimension of queries (100 clusters) to the private dataset, resulting in the efficiency of privacy costs.
>
> | $\sigma$ | 5 | 10 | 25 | 40 | 55 | 70 |
> | :---------- | :----| :--- | :-- | :-- | :-- | :-- |
> | Test Acc. | 85.2 (0.3) | 85.0 (0.2) | 84.5 (0.5) | 84.1 (0.1) | 83.8 (0.5) | 83.7 (0.1) |
> | Privacy Cost ($\epsilon$) | 1.23 | 0.58 | 0.22 | 0.14 | 0.10 | 0.08 |
>
> We hope these new results have addressed your concerns.

---

> ### Author Response · Authors · 2022-08-02
> **[2/2] Responses on the questions raised by Reviewer YJH7**
>
> > **Q1**: Do the authors have any insight as to why the random baseline outperforms the authors’ method in some cases? This seems counterintuitive especially since this baseline is perfectly privacy-preserving, and by using random sampling one is polluting the training dataset with out-of-distribution samples.
>
> **QA1**: We only notice one case where the random sampling significantly outperforms our method. That is the Clipart in adaptive model compression with 1000 budgets. Though random sampling is perfect in the case with no privacy cost and high accuracy, the method generally does not work well in all other cases. Therefore, it is not a stable choice in practice.
> For the counterintuitive result, we conjecture the possible reason is that the teacher model is not well trained on the Clipart domain. Thus, proximally sampled data causes inaccurate supervision from the teacher.
>
> > **Q2**: Does the K-center baseline require uploading the client dataset (or a transformed version of it) to the server? If so, this should be clarified since Tables 2-4 suggest that it is perfectly privacy-preserving.
>
> **QA2**: Neither K-center nor our method uploads client data to the server. Though our method upload some information to the cloud, the privacy risks are minor and accountable. We clarified the points on Page 6 of the revision.
>
> > **Q3**: For DomainNet, roughly what portion of the selected proximal data corresponds to the same domain as the client dataset? This could show the effectiveness of the sampling method in picking in-distribution samples.
>
> **QA3**: We are glad that you mentioned the question, which has been studied in Appendix B.2. By explicitly including a portion of in-distribution data (which are excluded from the client dataset), we measure the ratio of these data to be selected by ECOS, i.e., in-distribution true positive rate (ID TPR). In both Figure 4 and 5, we observe that our method can significantly increase the ID TPR compared to random selection (red line which represents the inherent ratio of ID data in the cloud dataset).

---

> ### Author Response · Authors · 2022-08-07
> **Thanks to Reviewer YJH7**
>
> Thanks again for the valuable comments. Especially, we appreciate your positive comments on the importance, rigor, and novelty of our work.
>
> We have now clarified the significance of our baselines, the sensitivity of privacy costs, and also show the effectiveness of our method in sampling ID data, as well as the new empirical results on the sensitivity of privacy costs and local training. Note that more detailed information is shown in our rebuttal summary.
>
> Please kindly let us know if anything remains unclear in our paper. We truly appreciate this opportunity to refine our work and shall be most grateful for any feedback you could give to us.

---

### Official Review · Reviewer_tiVh · 2022-07-12

**Rating:** 8
**Confidence:** 4
**Soundness:** 3 good
**Presentation:** 3 good
**Contribution:** 4 excellent

**Summary:**

Facing the importance of privacy in machine learning, the paper proposes an efficient and private outsourcing strategy to find surrogate datasets for training classifiers to be adapted to private datasets. The authors identify the limitation of traditional outsourcing strategies in the iid assumption of open-source data or inefficient or non-private data sharing. In addition, they reveal the challenges of outsourcing with non-iid open-source data and therefore propose a private, efficient and compelling solution regardless of the availability of labels in the open-source datasets. Extensive and various experiments demonstrate the effectiveness and flexibility of the proposed methods in practice.

**Questions:**

Can the proposed method be extended to other data domains, e.g., language models?

**Limitations:**

No obvious ethic issues in the paper. The authors have adequately discussed the limitation of the methods, like the data processing. Though more discussions on automating the data processing could be appreciated, it is an acceptable limitation within the scope of the paper.

**Strengths And Weaknesses:**

Strengths
1. (Novelty) To mitigate the high cost of model training, traditional outsourcing methods tends to transfer the local training to public data that are identically distributed to the local dataset. The idea of outsourcing training without uploading reference private data is quite novel with the relaxed non-iid assumption on the public dataset and provides an interesting direction for outsourcing that is both efficient and private.
2. (Impact) The paper is well motivated by the non-iid nature of open-source data and focuses on improving proximity, diversity and efficiency. Such practical considerations and goals make the method quite applicable in a wide range of real scenarios (with or without labels, label supervisions or model supervisions), and are promised with a high impact on privacy-preserving learning and efficient training outsourcing.
3. (Methodology) The proposed method is intuitive and is clearly guided by the importance of proximity, diversity and efficiency in improving the quality of selected open-source data. The intuition naturally leads to the principled objectives in sampling. Later experiments also demonstrate the importance of the three components. The privacy and efficiency (in terms of communication and computation) are solidly studied by theoretical analysis.
4. (Technical Quality) The paper is technically sound presenting significant improvements in privacy, computation and communication efficiency.
 - The experiments are extensively executed in various scenarios with different supervisions and the proposed method exhibits outstanding performance in various benchmarks. Due to the high cost of labels in practice, the authors clarifies that it is usually unrealistic to assume the availability of labels in open-source datasets. In the well-designed experiments, the authors provide comprehensive evaluations regarding the forms of training supervisions. It is shown that the method is flexible in multiple scenarios, including the sample selection for reducing the cost of manual labeling, adaptive model compression on calibrated open-source data distributions, and automated client labeling with improved accuracy.
 - In addition, the qualitative studies are quite helpful for understanding how principled objectives can lead to better performance. The visualization in Fig 3 provides another interesting perspective of the success, as well. The effects of important parameters like compression size and score scale s are well studied.
 - The privacy and efficiency of the method are evaluated not only theoretically but also empirically, which are quite solid.
 - Details of the implementation, including theorem proofs, datasets, and network architectures, are provided in detail meriting the reproducibility.

Weaknesses:
1. The experiments are limited to the domain of image recognition. The method could be considered in more general data assumptions.
2. The measurement of privacy could be described in detail early rather than in the appendix, for the convenience of readers who are not familiar with the notion of differential privacy.

---

> ### Author Response · Authors · 2022-08-02
> **Responses to Reviewer tiVh**
>
> We appreciate that you acknowledge the novelty, technical quality, and importance of our work. We hope our clarification can address your concerns.
>
> > Q: Can the proposed method be extended to other data domains, e.g., language models?
>
> A: We have good faith that our method can be applied in language models since we do not make any assumptions about data format. The proposed ECOS method relies on a well-learned representation space for recognizing similar distributions. With the advance of representation learning, modern language models can provide effective and distinguishable representations to benefit the effectiveness of ECOS in practice.

---

### Official Review · Reviewer_PrwV · 2022-07-16

**Rating:** 6
**Confidence:** 3
**Soundness:** 3 good
**Presentation:** 4 excellent
**Contribution:** 3 good

**Summary:**

This paper proposes a method of outsourcing client training in a privacy-preserving manner by effectively building and utilizing open-source data in the cloud. Problems with data heterogeneity is tackled by using a collaborative sampling solution. A series of experiments demonstrate the performance of this method.

**Questions:**

- Could ECOS be applied to a Federated Learning scenario, where multiple clients can learn from the proximal/diverse data on the cloud?

**Limitations:**

- No theoretical proofs on the convergence of models using ECOS.

**Strengths And Weaknesses:**

**Strengths**
- The proposed method appears to perform well under multiple learning scenarios
- Arguments on privacy show that ECOS could provide a viable method for training models in a safe manner.
- Extensive experiments demonstrate ECOS provides empirically well compared to other methods.

**Weaknesses**
- See *limitations* below

---

> ### Author Response · Authors · 2022-08-02
> **Responses on the questions raised by Reviewer PrwV**
>
> We appreciate that you praise the value, privacy evaluation, and empirical advantage of our work. We hope our responses below can address your major concerns.
>
> > **Q1**: Could ECOS be applied to a Federated Learning scenario, where multiple clients can learn from the proximal/diverse data on the cloud?
>
> **A1**: Yes. ECOS is applicable to federated learning. Since ECOS does not require local training, ECOS can be ad-hocly plugged into FL to obtain an auxiliary open-source dataset for enhancing the training per client. The ECOS can be used before federated training (when a pre-trained model is required) or during federated training (when the pre-trained model can be replaced by the on-training global model).
>
> > **Q2**: No theoretical proofs on the convergence of models using ECOS.
>
> **A2**: We agree that theoretical proofs could strengthen our work, but we want to clarify that the relationship between the EOCS and the convergence. In our paper, ECOS provides an efficient and principled solution to sample from open-source data. Then, models are trained on the sampled subset, where the convergence mainly relies on the post-hoc optimization algorithms. Various experiments in our paper demonstrated that the ECOS-sampled data can lead to better accuracy within the same iterations as other baselines.

---

> > ### Comment · Reviewer_PrwV · 2022-08-09
> > **Response to Authors**
> >
> > Thank you for your detailed answers to my questions. My concerns have been answered and will be keeping my positive score. I will look forward to the convergence analysis in the summary.

---

> > > ### Author Response · Authors · 2022-08-10
> > > **Thanks**
> > >
> > > Thanks for the positive responses. We will add the convergence analysis in our final version.
> > >
> > > In brief, based on [A], the convergence bound on the client loss can be informally derived  as
> > >
> > > $L(f_{\theta_T}, \mathcal{D}_t) \le (1 - \frac{\mu}{m})^T \epsilon(\theta_0, D_s) + {1\over 2} \Delta_H (\mathcal{D}_s, \mathcal{D}_t) + 4 \sqrt{2d \log (2 N) + \log(4/p) \over N}$
> > >
> > > where the first term is the convergence bound by $\mu$-strongly-convex and $m$-smooth assumptions, the second term is the divergence between the client ($D_t$) and the cloud ($D_s$) data distributions, and the third term describes the generalization error by the sample size. The bound provides additional insights on how the quality (the 2nd term) and the size (the 3rd term) of the subsampled cloud dataset affect the model performance on the client dataset.
> > >
> > > [A] J. Blitzer, K. Crammer, A. Kulesza, F. Pereira, and J. Wortman. Learning bounds for domain adaptation. Advances in neural information processing systems, 20, 2007

---

> ### Author Response · Authors · 2022-08-07
> **Thanks to Reviewer PrwV**
>
> We would like to thank the reviewer for taking the time to review our paper and for the valuable comments.
>
> Kindly let us know whether we have adequately addressed your comments on the applicability of our methods with federated learning and the convergence analysis. Note that more detailed information is provided in our rebuttal summary.
>
> We truly appreciate this opportunity to improve our work and shall be most grateful for any additional feedback you could give to us.

---

### Official Review · Reviewer_5x3V · 2022-07-25

**Rating:** 5
**Confidence:** 3
**Soundness:** 2 fair
**Presentation:** 2 fair
**Contribution:** 2 fair

**Summary:**

The work proposes to outsource model training from low-power and cost-effective end devices to powerful and cloud-based servers, where no access to the private data is given and instead a surrogate dataset is sampled from public data to obtain a proxy dataset. The sampling process was designed to be communication and computation efficient.

In the ECOS (Efficient Collaborative Open-source/world Sampling - there is a disagreement what ECOS stands for - compare lines 9 and 50) framework, the low-power clients do not send their private data but share features of their data that are used by the high-power server to adaptively find proximal public data for training the model (as a service). This facilitates the outsourcing of the model training without sharing local data by clients.

It is argued that public data have low sample efficiency in training the model that could be useful on a client's local data. This is due to the fact of including OOD (Out-Of-Distribution) samples from the public domain that degrade the accuracy on the client's data distribution.



**Questions:**

- It is very surprising in Table 5 that the random sampling performs better than the proximity objective. Could the proximity part from Equation 2 be further improved?

- How is the privacy cost computed in Table 4. How is the privacy loss from returning the coverage scores combined with the privacy loss incurred by labeling the proximal data with Private knn?

- Lack of future work or more potential methods. What are the other possible approaches to find which data the server should train the model? Wouldn't it be more practical if a client would point the server to the public data samples that should be used by the server to train the model? These can be also public datasets with labeled examples. The client could simply send links to the public data samples so the communication cost would be limited to the minimum. It seems that the best way to compare images is by applying a self-supervised model and then by comparing the representations from the private set with the representations from the public images.



**Limitations:**

The work assumes that there exist public data samples similar to the client's private data samples.

**Strengths And Weaknesses:**

Strengths:
- The authors tackle the important problem of outsourcing model training, which is of practical use. They incorporate the framework of differential privacy to account for the privacy leakage when a client instructs the server on the selection of the proximal dataset.

Weaknesses:
- Regarding the related work, the idea to "generate client-approximated labels for unlabeled public data, on which a model can be trained" was first proposed in https://arxiv.org/abs/1610.05755 and not in Private kNN.
- The definitions of proximity and diversity are omitted but this hinders the understanding of the paper. One has to refer to the related work [19] to find out the corresponding definitions - they should be included in this paper as they are the core of the method.
- In Tables 2 and 3, the Random method outperforms K-Center in all but 1 case. Isn't there any better baseline to compare with?
- It is not clear how the adaptive model compression works (e.g., presented as subplot (b) in Figure 2).


Typos:
- Line 154: the client uses
- Line 218: repeated three times with the following seeds {1,2,3}
- Line 224: three practical use cases
- Line 241: given higher budgets (not more)
- Line 316: ECOS only needs

---

> ### Author Response · Authors · 2022-08-02
> **[1/2] Responses on the weakness raised by Reviewer 5x3V**
>
> Thank you for acknowledging the importance and practical value of the studied problem and we also appreciate the detailed reviews. We tried our best and hope our clarifications can address your concerns.
>
> > Weakness
> >
> > **W1**: The idea to "generate client-approximated labels for unlabeled public data, on which a model can be trained" was first proposed in https://arxiv.org/abs/1610.05755 and not in Private kNN.
>
> **WA1**: Thanks for the suggestion. In our revision, we discussed the connection to the mentioned paper (namely, PATE), which was highlighted in Section 2. We hereby quote the original content in our paper below.
>
> “PATE and its variants were proposed to generate client-approximated labels for unlabeled public data, on which a model can be trained [29,30]. Without training multiple models by clients, Private kNN was a more efficient alternative that explored the private neighborhood of public images for labeling [43].”
>
> > **W2**: The definitions of proximity and diversity are omitted but this hinders the understanding of the paper.
>
> **WA2**: Thanks for the comment. For clarity, we have included the exact formulation of proximity and diversity in our revision, after Eq. (1).
>
> > **W3**: In Tables 2 and 3, the Random method outperforms K-Center in all but 1 case. Isn't there any better baseline to compare with?
>
> **WA3**: We agree that K-Center performs poorly in many cases, especially when the sampling budget is small. However, we argue that K-Center outperforms the random sampling in many cases. For example, SVHN, Clipart, Infograph, Painting, Real with 5000 budgets in Table 2, Infograph, Real and Sketch with 5000 budgets  in Table 3. The reason why K-Center looks like a poor solution in terms of the average accuracy is that K-Center suffers from catastrophic failures (over 20% accuracy drops versus other methods) in several cases, e.g., Quickdraw with 5000 budget in Table 2 and 3. These failures greatly degrade the average performance. Other than the failure cases, the K-Center could be a reasonable baseline in our benchmarks.
>
> > **W4**: It is not clear how the adaptive model compression works (e.g., presented as subplot (b) in Figure 2).
>
> **WA4**: The model compression is done by distilling a large model into a smaller one on selected samples. The large model is first pre-trained on all the open-source data, which data are distributed diversely and differently from the client data. To compress the model adaptable for the client, we leverage ECOS to sample a distributionally-similar subset from the open-source dataset for distillation.

---

> > ### Comment · Reviewer_5x3V · 2022-08-09
> > **Related work, definitions, K-Center, compression**
> >
> > Thank you for your answer, correcting the related work, adding the definitions, and clarifying the distillation process.
> >
> > Regarding Tables 2 and 3, the Random method outperforms K-Center in all but 1 case according to the Average Acc (%).

---

> > > ### Author Response · Authors · 2022-08-09
> > > **Response on the performance of K-Center**
> > >
> > > Thanks for your response. We also appreciate that you clarify your concerns in **W3**.
> > >
> > > As we argued in our rebuttal, we agree that the performance of K-Center is poor in terms of average accuracy. But the failure is, as we argued, attributed to some catastrophic failures (over 20% accuracy drops versus other methods), e.g., Quickdraw with 5000 budget in Table 2 and 3, that significantly degrade the average performance. Other than the failure cases, the K-Center outperforms the baselines in most cases. For example, in 12 out of 22 cases, K-Center outperforms Random in Table 2. In comparison, only 4 cases present when K-Center are worse than Random. In Table 3, K-Center performs worse in 3 out of 11 cases than Random while K-Center is better or comparable to Random in the other 8 cases. Moreover, beyond the two concerning tables, Table 4 shows that K-Center is an outstanding method in the hardest application where labels have to be inferred privately and possibly inaccurately by clients.
> > >
> > > The experimental results are more likely to show that K-Center is not a stable algorithm but can work well if the diversity overwhelms the optimality. When supervision is not available (e.g., in Table 4), diversity is more important to avoid learning from similar mistakes since two similar samples tend to augment the wrong supervision on models by similarly client-mislabeling.

---

> ### Author Response · Authors · 2022-08-02
> **[2/2] Responses on the questions raised by Reviewer 5x3V**
>
> > **Q1**: It is very surprising in Table 5 that the random sampling performs better than the proximity objective. Could the proximity part from Equation 2 be further improved?
>
> **QA1**: We are glad that you raised the concern. Since we spotted the same problem and discussed in Section 4.2 of the original version.
> The failure can be attributed to the nature of clustering that will include more similar and even redundant samples in each cluster. Simply sampling data from the proximity-ordered clusters will introduce probably class-imbalanced samples into the final set, resulting in worse generalization to the target domain.
>
> Therefore, the proximity term is more effective when the sampling diversity is promoted, which has been demonstrated in Table 5: the proximity strategy enhances the diversity-regularized method.
>
> > **Q2**: How is the privacy cost computed in Table 4. How is the privacy loss from returning the coverage scores combined with the privacy loss incurred by labeling the proximal data with Private knn?
>
> **QA2**: To compose the privacy costs from the two steps, we utilize the analytical moment accountant technique to get tight privacy bound [38], following the strategy used in private kNN for a fair comparison. We also clarified this point on Page 8.
>
> > **Q3**: Lack of future work or more potential methods. What are the other possible approaches to find which data the server should train the model? Wouldn't it be more practical if a client would point the server to the public data samples that should be used by the server to train the model? These can be also public datasets with labeled examples. The client could simply send links to the public data samples so the communication cost would be limited to the minimum. It seems that the best way to compare images is by applying a self-supervised model and then by comparing the representations from the private set with the representations from the public images.
>
> **QA3**: We appreciate the very valuable suggestions.
>
> (1) First, very similar to what is suggested, we compare the representations of the private and public datasets as shown in Line 3-4 of Algorithm 1. Then the client will score the clusters by measured proximity between public and private datasets. On receiving the cluster scores, the cloud will filter samples accordingly. The strategy can also be interpreted as: ‘client pointing the public samples for the server’ *by scoring public representations*.
>
> (2) Second, we argue that it is less efficient to send links of the images to the cloud than what it may sound. Since the clients do **not** have public images locally and need to first download them, for example, from the Internet. Compared to our method, the suggested method induces the extra communication costs on image downloading and the extra computation costs on feature extraction of public images by the client. In our method, we transfer the costs to the cloud, where the cloud will collect or download the images from the Internet, and extract low-dimensional features for sharing with the client. Thus, the costs on the client end can be greatly reduced.
>
> (3) The suggestion of using self-supervised models and comparing representations is an interesting idea and is very close to the proposed ECOS algorithm, except that we did not use self-supervised models but ImageNet pre-trained models. The reason is that ImageNet pre-trained models are widely available online. Using the off-the-self pre-trained models can also reduce the computation expense on the cloud since most cloud training is not free for the client. Self-supervised learning is a plausible alternative to such off-the-shelf models. We did not adopt the method since ImageNet pre-trained models perform better than the self-supervised ones in our DomainNet experiments. The gap may be resulted from that the used DomainNet dataset is smaller than the ImageNet.

---

> ### Author Response · Authors · 2022-08-07
> **Thanks to Reviewer 5x3V and follow-up message**
>
> We would like to thank the reviewer for the detailed comments, and particularly, for admitting our work with an important topic and practical motivation.
>
> We hope our response has adequately addressed your concerns regarding the effect of the proximity term, privacy accountant, and potential alternatives to our methods. Note that more information can be found in our rebuttal summary.
>
> Kindly let us know if anything is unclear. We truly appreciate your valuable feedback and comments that help us further improve our work.

---

> ### Author Response · Authors · 2022-08-09
> **A gentle reminder for response**
>
> Dear reviewer 5x3V, we deeply appreciate your initial feedback. We genuinely hope you could have a look at our clarifications and kindly let us know if anything is still unclear before the rebuttal ends. Your further feedback means a lot to us.

---

### Meta-Review · Area_Chair_SbwH · 2022-08-29

**Recommendation:** Accept
**Confidence:** Certain

**Metareview:**

This paper presents a novel framework for outsourcing the model training to cloud servers without the need for the clients to upload their data to the cloud. Unlike federated learning, the clients don't even need to perform any local training. In particular, the server relies on a large amount of open-source data to perform model training. To reduce the negative impact of out-of-domain (OOD) open-source data, the server performs **efficient collaborative open-source sampling** (ECOS) with the help of the client. The authors show that ECOS can be performed with a small amount of communication from client to server. Moreover, this communication can be made differentially private, as shown in the paper.  Overall, the proposed framework is novel and interesting. Its efficacy is demonstrated on two vision tasks of digit recognition and object recognition. The proposed framework outperforms a couple of natural baselines.

Most of the reviewers are positive about this work. Some of the reviewers had asked for some clarification which the authors adequately provided during the rebuttal. Notably, the authors provided a generalization bound,  results on another baseline (where clients train the model locally), and accuracy vs. differential privacy cost experiments. The authors are encouraged to include all these results in the revised paper.

Even though the presentation of the paper has significantly improved after the author-reviewer discussion, there is still scope for some improvements: 1) Description of the sampling objective in Section 3.2 lacks flow. It's not clear why $\hat{S}$ is introduced. $\hat{S}$ does not even feature in the description of Algorithm 1. Please address this. 2) $\hat{D}^q$ is used to denote the $R$ centroids of the public data, whereas $\hat{D}^p$ (line 4, Algorithm 1) is used to denote the features of client data. Please use consistent notations.

Even though the authors have mentioned that their framework can potentially work for language models with minimal modifications (e.g., using appropriate representations), it needs to be verified via detailed experiments. The authors should either include results on this or add this as a potential avenue for future work.

**Award:**

No

---

### Decision · Program_Chairs · 2022-09-14

Accept